# PAMA: Dual-Memory Augmentation Assisted Pseudo-Anomaly Contrastive Learning for Multivariate Time Series Anomaly Detection

## Abstract

For multivariate time series anomaly detection, most methods assume that training data are clean and ignore the characteristics of anomalous data. They often suffer from the overgeneralization problem during the reconstruction process. To address these problems, dual-memory augmentation assisted pseudo-anomaly contrastive learning for multivariate time series anomaly detection (shorted as PAMA) is proposed. First, the prior knowledge of anomalies is utilized to generate pseudo anomalies from the original time series. The normal and pseudo-anomalous feature representations that contain global and local information are respectively achieved by the global and local encoders. Two independent memory modules are constructed to further memorize normal and pseudo-anomalous prototypes. Second, a dual-memory augmentation mechanism is proposed to conduct data augmentation upon the normal and pseudo-anomalous feature representations and then obtain the memory-augmented feature representations. Third, the pseudo-anomaly contrastive learning is proposed to perform temporal contrastive learning and instance contrastive learning on the obtained memory-augmented representations. Compared with the fourteen baseline methods, the experimental results demonstrate that PAMA achieves the optimal detection performance.

## 1 Introduction

As an important branch of data analysis, multivariate time series anomaly detection (MTSAD) aims to identify and analyze non-typical or anomalous patterns that deviate from normal patterns. Unfortunately, detecting anomalies from large amount of complex multivariate time series is challenging. On the one hand, anomalies are complex and possess various types in real-world scenarios, such as global point anomalies, contextual point anomalies, and pattern anomalies. On the other hand, noise often exist in multivariate time series composed of normal data. As is well known, there are two types of noise, i.e., feature noise and label noise. In this study, we only focus on feature noise.

Existing methods often get into trouble for correctly classifying noise and anomalous data. Mainstream methods, whether based on autoencoders, contrastive learning, or others, are mostly constructed by relying on the assumption of single normality or multiple normalities. Their core logic is to assume that training set contains almost no anomalous data, and anomalies are identified solely by learning the feature patterns of normal data. For MTSAD, MEMTO (Song et al., 2023), U-Transformer (Qin et al., 2023), and MSDMM (Xue et al., 2024) utilize memory terms during the training procedure to learn normal patterns and reconstruct time series representations. However, in practical scenarios, training data are often mixed with various types of noise and a small number of anomalous points. These interferences severely hinder the detection models to learn normal patterns. To address this issue, some methods (e.g., CutAddPaste (Wang et al.,

2024)) leverage prior knowledge of anomalies to generate pseudo anomalies for model training. Inspired by these methods, we design a pseudo-anomalous memory module that dynamically stores and updates representative pseudo-anomalous prototypes to effectively capture anomalous patterns. As shown in Fig. 1, the above-mentioned three conventional methods only learn normal patterns but cannot learn anomalous patterns during the training process. The main reason is that their memory modules can only learn from normal data, so that only normal prototypes can be stored. In contrast, our memory module consider both normal and pseudo-anomalous data. Hence, both normal and anomalous patterns can be obtained.

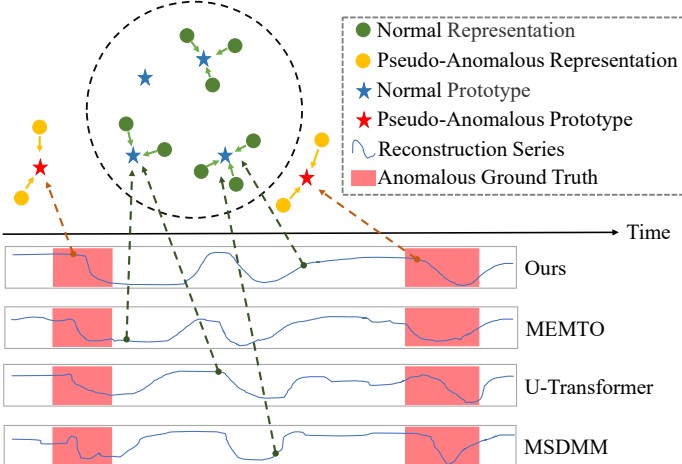

Figure 1: The different learning objectives of the conventional methods and our proposed method. Dots of different colors represent different feature representations, and five-pointed stars of different colors represent different prototypes learned during the training process.

To better learn and utilize anomalous patterns, we propose dual-memory augmentation assisted pseudo-anomaly contrastive learning for MTSAD, abbreviated as PAMA. It utilizes prior knowledge of anomalies to generate pseudo anomalies to simulate anomalies in time series. Furthermore, PAMA uses a dual-memory augmentation mechanism to perform data augmentation on normal representations and pseudo-anomalous representations separately. In pseudo-anomaly contrastive learning, temporal contrastive learning and instance contrastive learning are organized to learn normal and anomalous patterns from different memory-augmented feature representations. Finally, in uncertainty learning, the regularization loss and distance loss are used to distinguish between normal and pseudo-anomalous representations, thereby reducing noise interference.

The contributions of the study are summarized as follows.

- We design a dual-memory augmentation (DMA) mechanism to get memory-augmented representations by performing data augmentation through the normal and pseudo-anomalous memory modules, respectively. DMA can effectively enhance the learning ability for prototype features by performing augmentation processing on different representations.

- We propose pseudo-anomaly contrastive learning (PACL), which performs temporal contrastive learning and instance contrastive learning to obtain reliable and stable normal prototypes together with complex and diverse pseudo-anomalous prototypes. Moreover, PACL can efficiently improve the detection performance on anomalies.

- We modified the conventional pseudo-anomaly generation (PAG) module to make full use of seasonality and trend of time series to generate pseudo anomalies that simulate the actual anomalous situation.

## 2 RELATED WORK

MTSAD is an important research direction in time series analysis, with widespread applications in industrial monitoring, financial risk control, medical diagnosis, and other fields (Darban et al., 2024). To capture the normal patterns of multivariate time series (MTS), OmniAnomaly (Su et al., 2019) uses stochastic recurrent neural networks to learn robust representations. InterFusion (Li et al., 2021) utilizes two random hidden variables to model normal patterns in MTS via a hierarchical variational autoencoder. TFAD (Zhang et al., 2022) uses a variety of data augmentation mechanisms to perform frequency domain and time domain analysis simultaneously through a time series decomposition module. Since Transformer can effectively model long-term dependencies in time series, transformer-based reconstruction methods have been widely used in MTSAD (Vaswani et al., 2017). Anomaly Transformer improves the distinguiability of normal and anomalous data by amplifying association discrepancy through the max-min strategy (Xu et al., 2022). VTT (Kang & Kang, 2024) learns temporal dependencies through temporal self-attention and variable correlations via variable self-attention to fulfill anomaly detection. However, the aforementioned detection methods mainly focus on learning normal patterns but completely neglect to extract anomalous patterns. They are difficult to tackle anomalies with small differences from normal data in complex scenarios.

The task of an MTSAD method based on contrastive learning is to maximize the similarity between normal data while minimizing the similarity between normal and anomalous data. TimeAutoAD (Jiao et al., 2022) adopts three different data augmentation strategies to generate pseudo negative time series and utilizes self-supervised contrastive loss to distinguish the original time series from the pseudo negative time series. CAE-AD (Zhou et al., 2022) performs time-domain data augmentation and frequency-domain data augmentation, while extracting normal patterns and learning local invariant features by multi-granularity contrastive learning. Nevertheless, the negative samples in the above-mentioned contrastive learning methods are all derived from the augmented normal data, which makes these methods difficult to learn anomalous patterns.

Recently, synthetic anomalies have been incorporated in the the training procedure of MTSAD (Carmona et al., 2022). CutAddPaste (Wang et al., 2024) converts previously known knowledge of anomalies into a data augmentation strategy based on the anomaly assumption. By generating representative pseudo anomalies and combining them with the original data for training, its detection performance is greatly improved. However, the above-mentioned method simply disrupts the local sequence, ignoring the inherent seasonality and long-term trend of the time series, and only generate simple point anomalies and local trend anomalies, resulting in a significant difference between the pseudo-anomalous data generated by the foresaid method and the actual anomalies. Therefore, these methods have difficulty learning anomalous patterns during the training process.

## 3 METHODOLOGY

Let an MTS with length $S$ be divided into multiple window-level time series $\mathbf{X} = \{\mathbf{x}_1, \ldots, \mathbf{x}_T\}$ through a sliding window with size $T$. Each data $\mathbf{x}_t \in \mathbb{R}^C$ ($t = 1, \ldots, T$) is collected from an industrial sensor or machine at a certain time step. The problem can be described as follows. Given a training series $\mathbf{X}$, for another unknown test series $\mathbf{X}_{\text{test}} = \{\mathbf{x}_t\}_{t=1}^L$ with length $L$, an MTSAD model calculates an anomaly score $s_t$ to substitute its corresponding binary class label. The predicted labels $\mathbf{Y}_{\text{test}} = \{y_1, \ldots, y_L\} \in \{0, 1\}$ can be obtained by comparing $s_t$ with a predefined threshold $\xi$, where 1 denotes anomaly and 0 denotes normal. In this section, the proposed PAMA is described in detail. Its model structure is shown in Fig. 2.

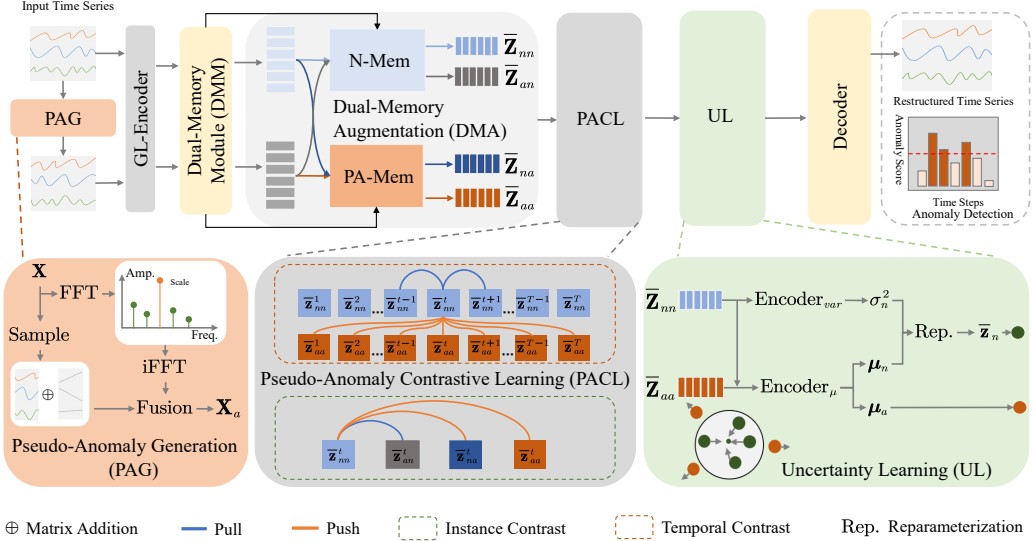

Figure 2: Model structure of PAMA.

## 3.1 PSEUDO-ANOMALY GENERATION

The PAG module for generating pseudo anomalies in PAMA is an innovative approach based on anomaly assumptions, inspired by CutAddPaste (Wang et al., 2024). The aim of PAG is to combine the prior knowledge of anomalies and use data augmentation techniques to generate pseudo anomalies that contain trend perturbations and seasonal perturbations.

PAG performs random sampling on the original time series $\mathbf{X} \in \mathbb{R}^{T \times C}$. Its sampling position, sampling rate and sampling length are respectively set to be $p_s$, $r_s$ and $l_s = r_s \times T$ with $0 \le p_s \le T - l_s$. The sampled time series can be obtained as $\mathbf{X}_s \in \mathbb{R}^{l_s \times C}$. PAG generates a random trend perturbation $\mathbf{T}_p \in \mathbb{R}^{l_s \times C}$. For each channel $\mathbf{T}_p^i$, where $i \in \{1, \ldots, C\}$, the direction $d_i$ and amplitude $a_i$ are randomly selected. Let $\mathbf{T}_p^i(t)$ be the linear perturbation at the time step $t$, which is given by

$$\mathbf{T}_p^i(t) = t \frac{a_i}{l_s - 1} d_i. \tag{1}$$

The trend perturbation series $\mathbf{X}_{\text{tp}} \in \mathbb{R}^{l_s \times C}$ can thus be obtained by adding $\mathbf{T}_p$ and $\mathbf{X}_s$. Meanwhile, the original time series is converted to the frequency domain by FFT. Furthermore, PAG calculates the amplitude of each frequency component for all features in the frequency domain and selects the frequency component $f_i^{\text{max}}$ with the largest amplitude on the $i$-th feature. $\mathbf{X}(f_i^{\text{max}})$ is then scaled by the perturbation factor $\alpha$. Subsequently, PAG replaces the component in $\tilde{\mathbf{X}}(f)$ at frequency $f_i^{\text{max}}$ with the scaled component while keeping the remaining components unchanged, that is

$$\tilde{\mathbf{X}}(f) = \begin{cases} \alpha \cdot \mathbf{X}(f) & \text{if } f = f_{\text{max}}^i \\ \mathbf{X}(f) & \text{otherwise} \end{cases} \tag{2}$$

$\tilde{\mathbf{X}}$ is thus converted back to the time domain by iFFT to obtain $\mathbf{X}_{\text{sp}}$.

Finally, part of the final pseudo anomalies $\mathbf{X}_a$ with random position $p_g$ as the starting point and length $l_s$ is replaced by the trend perturbation series $\mathbf{X}_{\text{tp}}$, and the rest is obtained from the seasonal perturbation series $\mathbf{X}_{\text{sp}}$ of the corresponding time points. The process can be expressed as follows.

$$\mathbf{X}_a(i,j) = \begin{cases} \mathbf{X}_{\text{tp}}(i - p_g, j), & \text{if } p_g \leq i \leq p_g + l_s - 1; \\ \mathbf{X}_{\text{sp}}(i,j), & \text{Otherwise.} \end{cases} \tag{3}$$

## 3.2 DUAL-MEMORY AUGMENTATION

PAMA adopts the global-local encoder (GL-Encoder) of UR-DMU (Zhou et al., 2023). GL-Encoder uses Transformer to capture long-term trends and learn global dependencies, while utilizes convolutional neural network to capture short-term fluctuations and learn local dependencies. For ease of presentation, we denote the original series $\mathbf{X}$ as $\mathbf{X}_n$. PAMA simultaneously input $\mathbf{X}_n$ and $\mathbf{X}_a$ into the GL-Encoder, the normal representations $\mathbf{Z}_n$ and the pseudo-anomalous representations $\mathbf{Z}_a$ containing the global and local information can be obtained, respectively. We can get

$$\mathbf{Z}_p = \text{GL-Encoder}(\mathbf{X}_p), \tag{4}$$

where $p \in \{n, a\}$.

Inspired by MEMTO (Song et al., 2023), a dual-memory module is designed, which consists of a normal memory module $\mathbf{M}_n$ and a pseudo-anomalous memory module $\mathbf{M}_a$. The latter module can capture and learn pseudo-anomalous prototypes from various pseudo-anomalies. To fully utilize the prototypes in the dual-memory module, the DMA mechanism is designed. First, DMA calculates the similarity scores $\mathbf{S}_{pq} \in \mathbb{R}^{T \times N}$ between the memory module $\mathbf{M}_q \in \mathbb{R}^{N \times d_{\text{model}}}$ and the representations $\mathbf{Z}_p \in \mathbb{R}^{T \times d_{\text{model}}}$, where $q \in \{n, a\}$. Hence,

$$\mathbf{S}_{pq} = \text{Sigmoid}\left(\frac{\mathbf{Z}_p \mathbf{M}_q^{\mathsf{T}}}{\sqrt{d_{\text{model}}}}\right), \tag{5}$$

where $\text{Sigmoid}(\cdot)$ denotes the sigmoid activation function.

Then, the dual-memory module and the similarity scores are used to generate the memory-augmented representations. The DMA mechanism adopts two memory-augmentation modules, namely, N-MemM and PA-MemM, which store $\mathbf{M}_n$ and $\mathbf{M}_a$, respectively. N-MemM and PA-MemM take different representations as input and generate different memory-augmented representations. The memory-augmentation process is expressed as

$$\bar{\mathbf{Z}}_{pq} = \mathbf{S}_{pq}\mathbf{M}_q. \tag{6}$$

## 3.3 PSEUDO-ANOMALY CONTRASTIVE LEARNING

In temporal contrastive learning, the distance between the representations at two adjacent time steps is usually less than that between those at two non-adjacent time steps. Hence, for the time step $t$, the representations of $\bar{\mathbf{Z}}_{nn}$ at its two adjacent time steps, i.e., $t - 1$ and $t + 1$ are utilized as positive pairs. Moreover, the representation of $\bar{\mathbf{Z}}_{nn}$ at $t$ and any representation in $\bar{\mathbf{Z}}_{aa}$ are taken as negative pairs. The temporal contrastive loss is given by

$$\mathcal{L}_{\text{temp}}^{(i,t)} = -\log \frac{\exp\left(\bar{\mathbf{z}}_{nn}^{i,t} \cdot \bar{\mathbf{z}}_{nn}^{i,t-1}\right) + \exp\left(\bar{\mathbf{z}}_{nn}^{i,t} \cdot \bar{\mathbf{z}}_{nn}^{i,t+1}\right)}{\sum_{t' \in T} \exp\left(\bar{\mathbf{z}}_{nn}^{i,t} \cdot \bar{\mathbf{z}}_{aa}^{i,t'}\right)}. \tag{7}$$

In instance contrastive learning, the representations $\bar{\mathbf{Z}}_{an}$ are obtained by using the normal memory module to perform memory-augmentation on the pseudo-anomalous representations. Moreover, the representations $\bar{\mathbf{Z}}_{na}$ are obtained by using the pseudo-anomalous memory module to perform memory-augmentation on the normal representations. Furthermore, the representations of $\bar{\mathbf{Z}}_{nn}$ and $\bar{\mathbf{Z}}_{an}$ at the same time step are used as positive pairs, while the representations of $\bar{\mathbf{Z}}_{nn}$ and $\bar{\mathbf{Z}}_{na}$ at the same time step are utilized as negative pairs.

In the same way, the representations of $\bar{\mathbf{Z}}_{nn}$ and $\bar{\mathbf{Z}}_{aa}$ at the same time step are also taken as negative pairs. The instance contrastive loss can be expressed as

$$\mathcal{L}_{\text{inst}}^{(i,t)} = -\log \frac{\exp\left(\bar{\mathbf{z}}_{nn}^{i,t} \cdot \bar{\mathbf{z}}_{an}^{i,t}\right)}{\exp\left(\bar{\mathbf{z}}_{nn}^{i,t} \cdot \bar{\mathbf{z}}_{na}^{i,t}\right) + \exp\left(\bar{\mathbf{z}}_{nn}^{i,t} \cdot \bar{\mathbf{z}}_{aa}^{i,t}\right)}. \tag{8}$$

In summary, the total contrastive loss is defined as

$$\mathcal{L}_{\text{con}} = \frac{1}{TB} \sum_{t=1}^{T} \sum_{i=1}^{B} \left(\mathcal{L}_{\text{temp}}^{(i,t)} + \mathcal{L}_{\text{inst}}^{(i,t)}\right), \tag{9}$$

where $B$ represents the batch size.

### 3.4 UNCERTAINTY LEARNING

PAMA modifies the conventional uncertainty learning (UL) to effectively learn data uncertainty. Through $\text{Encoder}_\mu$ and $\text{Encoder}_{var}$, we can obtain the mean vector $\boldsymbol{\mu}_n^d$ and variance $(\sigma_n^d)^2$ for each variate of $\bar{\mathbf{Z}}_{nn}$. Moreover, $\boldsymbol{\mu}_a^d$ can also be achieved for each variate of $\bar{\mathbf{Z}}_{aa}$. UL constraints the $d$-th variate of $\bar{\mathbf{Z}}_{nn}$, i.e., $\bar{\mathbf{z}}_n^d$ to obey the normal distribution $\mathcal{N}\left(\boldsymbol{\mu}_n^d, \left(\sigma_n^d\right)^2 \mathbf{I}\right)$. Note that $\bar{\mathbf{z}}_n^d$ is generated by the reparameterization technique to ensure its differentiability. To suppress instability caused by excessive $\sigma_n^d$, $\mathcal{L}_{\text{kl}}$ is used as the regularization loss. The distance loss $\mathcal{L}_{\text{dis}}$ is adopted in order to maximize the distance between $\bar{\mathbf{z}}_n^d$ and $\bar{\mathbf{z}}_a^d$, and simultaneously minimize the distance between $\bar{\mathbf{z}}_n^d$ and its corresponding cluster center. Let the normal representations after UL be $\bar{\mathbf{Z}}_n$. $\bar{\mathbf{Z}}_n$ and $\mathbf{Z}_n$ are concatenated and then inputted into the decoder composed of linear layers. The reconstructed time series $\hat{\mathbf{X}}$ can be ultimately obtained as follows.

$$\hat{\mathbf{X}} = \text{Decoder}\left(\text{Concat}(\bar{\mathbf{Z}}_n, \mathbf{Z}_n)\right). \tag{10}$$

The complete description of UL is included in the appendix B.4.

### 3.5 LOSS FUNCTION AND ANOMALY SCORES

The reconstruction loss of PAMA is given by

$$\mathcal{L}_{\text{rec}} = \frac{1}{T} \sum_{t=1}^{T} \|\mathbf{x}_t - \hat{\mathbf{x}}_t\|_2^2. \tag{11}$$

The entropy loss $\mathcal{L}_{\text{entr}}$ of MEMTO (Song et al., 2023) is used as an auxiliary loss in the dual-memory module. During the training phase, the overall loss function of PAMA is expressed as

$$\mathcal{L} = \mathcal{L}_{\text{rec}} + \lambda_1 \mathcal{L}_{\text{entr}} + \lambda_2 \mathcal{L}_{\text{con}} + \lambda_3 \mathcal{L}_{\text{kl}} + \lambda_4 \mathcal{L}_{\text{dis}}. \tag{12}$$

To minimize the overall loss function $\mathcal{L}$, batches are selected for training, and the Adam optimizer is used to update the network weights for each training round.

During the testing phase, $\mathbf{X}_{\text{test}} = \{\mathbf{x}_t\}_{t=1}^L$ is inputted into PAMA and through the trained decoder. The reconstructed series $\hat{\mathbf{X}}_{\text{test}} = \{\hat{\mathbf{x}}_t\}_{t=1}^L$ can thus be obtained. To fully utilize the prototypes in the dual-memory module, the distance between each feature representation $\mathbf{z}_t$ in the latent space and its nearest prototype $\mathbf{m}_t^n$ in the normal memory module is calculated. Since each prototype in the normal memory module represents a normal pattern, the distance between anomalous representation and normal prototype is

greater than that between normal representation and normal prototype. The anomaly score $s_t$ is calculated as follows.

$$s_t = \|\mathbf{z}_t - \mathbf{m}_t^n\|^2 \times \|\mathbf{x}_t - \hat{\mathbf{x}}_t\|^2. \tag{13}$$

Given a predefined threshold $\xi$, if $s_t > \xi$, the time step $t$ is classified as anomaly. Otherwise, it is classified as normal. That is,

$$y_t = \begin{cases} 1, & s_t > \xi; \\ 0, & s_t \leq \xi. \end{cases} \tag{14}$$

The training and testing algorithms of PAMA are summarized in appendix B.6.

## 4 EXPERIMENTAL RESULTS

### 4.1 EXPERIMENTAL SETUP

**Datasets**. In the following experiments, five commonly used MTSAD benchmark datasets are used, including MSL, SMAP (Hundman et al., 2018), PSM (Abdulaal et al., 2021), SWaT (Goh et al., 2016), SMD (Su et al., 2019).

**Implementation Details** The training dataset is divided into 80% training set and 20% validation set. In the experiments, three commonly used metrics, namely, Precision (Pre), Recall (Rec) and F1-score (F1), are used to evaluate and compare the proposed PAMA with baseline methods. We use the commonly used point adjustment technique for comparison (Shen et al., 2020).

**Baseline Methods**. A series of experiment are conducted to compare PAMA with fourteen methods including ITAD (Shin et al., 2020), DAGMM (Zong et al., 2018), OmniAnomaly (Su et al., 2019), InterFusion (Li et al., 2021), MAD-GAN (Li et al., 2019), Anomaly Transformer (Xu et al., 2022), DCdetector (Yang et al., 2023), MEMTO (Song et al., 2023), U-Transformer (Qin et al., 2023), GSC-MAD (Zhang et al., 2024), CAE-AD (Zhou et al., 2022), MST-GAT (Ding et al., 2023), H-PAD(Shen, 2025) and CARLA(Darban et al., 2025).

More experimental details and additional experiments can be found in the appendix C and appendix D, respectively.

### 4.2 RESULTS AND ANALYSIS

Table 1: Testing results of different methods on the five datasets (%), where the term 'Avg.' refers to the average F1-score. The optimal results are highlighted in bold and the suboptimal results are underlined.

| Model | MSL | | | SMAP | | | PSM | | | SMD | | | SWaT | | | Avg |
|---|---|---|---|---|---|---|---|---|---|---|---|---|---|---|---|---|
| | Pre | Rec | F1 | Pre | Rec | F1 | Pre | Rec | F1 | Pre | Rec | F1 | Pre | Rec | F1 | F1 |
| DAGMM | 89.60 | 63.93 | 74.62 | 86.45 | 56.73 | 68.51 | 93.49 | 70.03 | 80.08 | 89.92 | 57.80 | 70.40 | 67.30 | 49.89 | 57.30 | 70.18 |
| MAD-GAN | 85.17 | 89.91 | 87.48 | 80.49 | 82.14 | 81.31 | - | - | - | **98.97** | 63.74 | 77.54 | 85.17 | 89.91 | 87.48 | 83.45 |
| OmniAnomaly | 89.02 | 86.37 | 87.67 | 92.49 | 81.99 | 86.92 | 88.39 | 74.46 | 80.83 | 81.42 | 84.30 | 82.83 | 83.68 | 86.82 | 85.22 | 84.69 |
| ITAD | 69.44 | 84.09 | 76.07 | 82.42 | 66.89 | 73.85 | 72.80 | 64.02 | 68.13 | 63.13 | 52.08 | 57.08 | 86.22 | 73.71 | 79.48 | 70.92 |
| InterFusion | 81.28 | 92.70 | 86.62 | 89.77 | 88.52 | 89.14 | 83.61 | 83.45 | 83.52 | 80.59 | 85.58 | 83.01 | 87.02 | 85.43 | 86.22 | 85.70 |
| MST-GAT | 95.06 | 89.10 | 91.98 | 91.26 | 89.83 | 90.54 | - | - | - | 98.73 | 72.41 | 83.55 | - | - | - | 88.69 |
| CAE-AD | 86.11 | 96.88 | 91.19 | 88.24 | 98.36 | 93.02 | - | - | - | - | - | - | 92.65 | 94.91 | 93.76 | 92.66 |
| Anomaly Trans | 91.92 | 96.03 | 93.59 | 93.59 | **99.41** | 96.41 | 96.91 | 98.90 | 97.89 | 90.98 | 93.93 | 92.41 | 88.41 | 92.58 | 90.40 | 94.20 |
| MEMTO | 92.07 | 96.76 | 94.36 | 93.75 | 99.29 | 96.44 | 97.44 | 98.96 | 98.18 | 92.89 | 95.13 | 94.00 | 89.13 | 97.76 | 93.25 | 95.24 |
| U-Transformer | **95.63** | 95.43 | 95.53 | **96.63** | 97.26 | 96.94 | 95.49 | 88.58 | 91.91 | 96.13 | 81.43 | 88.17 | 67.78 | 84.13 | 75.07 | 89.52 |
| DCdetector | 92.03 | **98.82** | 95.30 | 94.54 | 98.81 | 96.63 | 97.18 | 98.33 | 97.75 | 93.23 | **99.30** | 96.17 | 83.59 | 91.10 | 87.18 | 94.61 |
| GSC-MAD | 94.19 | 93.09 | 93.63 | 89.57 | 98.35 | 93.76 | 97.97 | 99.14 | 98.89 | 96.73 | 95.11 | 95.91 | 92.25 | 94.32 | 93.32 | 95.10 |
| H-PAD | 94.05 | 96.88 | 95.45 | 96.00 | 98.45 | 97.21 | 98.82 | **99.41** | **99.12** | 92.86 | 98.20 | 95.45 | **94.14** | **98.89** | **95.09** | 96.46 |
| **PAMA** | 94.34 | 97.93 | **96.10** | 95.95 | 98.95 | **97.42** | **98.86** | 99.35 | 99.11 | 92.05 | 99.29 | 95.53 | 92.44 | 97.36 | 94.84 | **96.60** |

Table 2: Testing results of different methods on the five datasets (%), where the term 'AR' refers to AUC-ROC and the term 'AP' refers to AUC-PR.

| Datasets | MSL | | SMAP | | PSM | | SWaT | | SMD | | Avg. | |
| Models | AR | AP | AR | AP | AR | AP | AR | AP | AR | AP | AR | AP |
|---|---|---|---|---|---|---|---|---|---|---|---|---|
| Anomaly Trans | 48.72 | 10.64 | 49.67 | 12.50 | 48.56 | 29.42 | 29.40 | 8.82 | 47.28 | 3.70 | 44.73 | 13.02 |
| MEMTO | 49.99 | 10.48 | **59.59** | 16.29 | 49.75 | 26.96 | 45.41 | 11.45 | 73.24 | 10.35 | 55.60 | 15.11 |
| DCdetector | 50.06 | 10.61 | 48.87 | 12.48 | 49.83 | 27.64 | 49.74 | 11.60 | 48.77 | 41.16 | 49.45 | 20.76 |
| H-PAD | **58.67** | 14.06 | 59.13 | 15.30 | **69.15** | **48.13** | 67.93 | 14.05 | **79.43** | **51.92** | **66.86** | **28.69** |
| CARLA | 50.20 | 13.50 | 54.20 | 14.85 | 43.25 | 24.40 | **70.70** | **30.10** | 45.20 | 15.10 | 52.71 | 19.59 |
| PAMA | 50.68 | **14.20** | 59.09 | **17.53** | 51.38 | 32.66 | 64.9 | 17.86 | 55.53 | 25.31 | 56.32 | 21.51 |

A series of numerical experiments are conducted to compare PAMA and thirteen baseline methods on the five benchmark datasets. As shown in Table 1, PAMA achieves superior detection performance on most benchmark datasets, with the average F1-score reaching the optimal level. In comparison with CAE-AD which is one of methods based on contrastive learning, PAMA significantly improves the F1-score on MSL, SMAP, and SWaT. The outperformance relies on its pseudo-anomaly contrastive learning based on the dual memory-augmented representations, considering the similarities not only between adjacent time steps but also highly correlated non-adjacent steps. Compared with reconstruction-based methods, the performance of PAMA is also improved. Furthermore, PAMA is also compared with recent methods on five datasets under AUC-ROC and AUC-PR in Table 2. PAMA exhibits higher accuracy and better generalization capability on anomaly datasets with different anomaly rates.

### 4.3 ABLATION STUDY

To conduct an in-depth analysis of PAMA components, ablation study is carried out on each component and experimental results are summarized in Fig. 3(a). Baseline refers to the introduction of pseudo anomalies into MEMTO. GL-Encoder, UL and PACL denote the model of the baseline model individually attaching with GL-Encoder, uncertainty learning and pseudo-anomaly contrastive learning correspondingly. Overall, PAMA performs best across all datasets, followed by the PACL model. Introducing UL and PACL significantly improves model performance, while PAMA achieves the highest F1 score.

Further experiments are carried out to explore the effectiveness of pseudo-anomaly contrastive learning for PAMA. In Fig. 3(b), w/o Instance-CL means that the full model removes instance contrastive learning, w/o temporal-CL means that the full model removes temporal contrastive learning and w/o PACL refers to no pseudo-anomaly contrastive learning used during training. The experimental results in Fig. 3(b) strongly demonstrate the significant role of the proposed pseudo-anomaly temporal contrastive learning and instance contrastive learning in improving model performance. Our conclusion is that PACL can effectively help the model better learn normal and anomalous patterns, thereby enhancing the ability of the model to distinguish between normal and anomalous data. Temporal contrastive learning uses the difference of similarity at different time steps to learn normal and anomalous temporal patterns, while instance contrastive learning learns different patterns by capturing the differences between different memory-augmented representations. Therefore, normal and anomalous patterns can be effectively learned using PACL.

### 4.4 PARAMETER ANALYSIS

To investigate the impact of different values of hyperparameters on the performance of PAMA, we conducted parameter sensitivity experiments on the perturbation factor $\alpha$ in Eq.(2) and the sampling ratio $r_s$. In real applications, $\alpha$ and $r_s$ are related to periodic and trend disturbances. Higher values of $\alpha$ and $r_s$ indicate stronger impact on sensor.

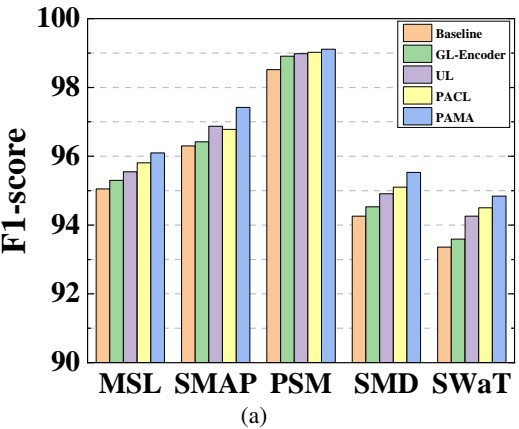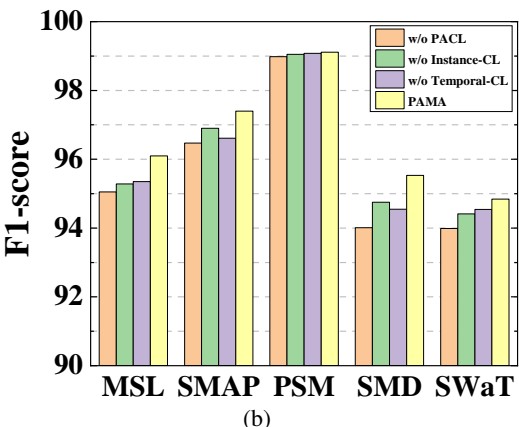

Figure 3: Ablation experiments on model performance by different modules and different contrastive learning in PAMA.

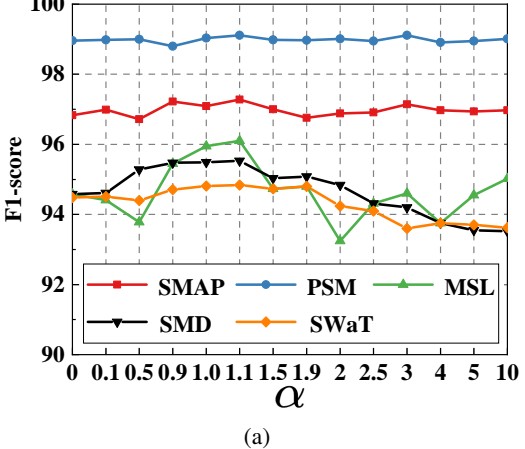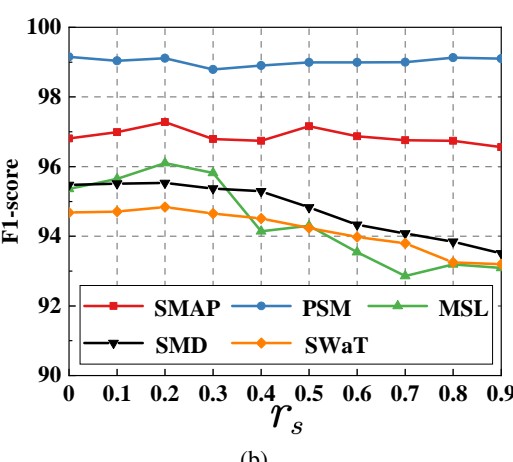

Figure 4: Influence of different values of hyperparameters in the pseudo-anomaly generation process on the performance of PAMA upon the three benchmark datasets.

As shown in Fig. 4(a), for $\alpha$ perturbation, F1-score of PAMA fluctuates significantly on MSL, while F1-score remains relatively stable on other datasets. When $\alpha$ is set to 1.1, PAMA achieves the optimal performance on all five datasets, indicating that the model can effectively handle mild periodic perturbations. For the sampling ratio $r_s$, it can be observed from Fig. 4(b) that F1-score of PAMA on MSL, SMD and SWaT shows a significant downward trend, while it decreases slowly on PSM and SMAP. The optimal performance is achieved for $r_s = 0.2$, indicating that PAMA can effectively handle small trend perturbations. By adjusting values of $\alpha$ and $r_s$, PAMA can learn periodic and trend changes which is crucial for learning anomalous data patterns and improving robustness.

## 5 CONCLUSION

In this study, a novel MTSAD method named PAMA is proposed. PAMA respectively utilizes normal memory module and pseudo-anomalous memory module to learn normal patterns and pseudo-anomalous patterns, employs dual-memory augmentation for data augmentation, and applies contrastive learning and uncertainty learning to leverage pseudo-anomalous patterns. Experimental results demonstrate that PAMA outperforms relevant baseline models on five datasets, with each component playing a crucial role to improve detection performance. Although PAMA demonstrates outstanding detection performance, it requires a large amount of time and storage costs due to the use of dual-memory module. In the future, we will explore new ways of data augmentation to utilize pseudo anomalies without using dual-memory modules.

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

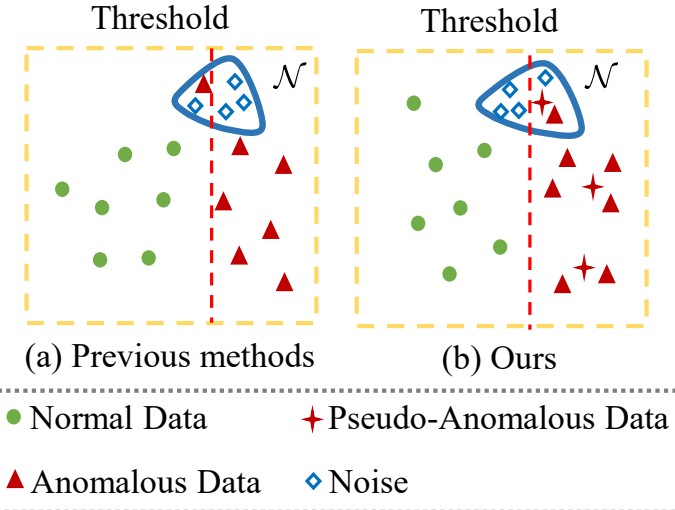

Figure 5: The different processing outcomes of the previous methods and our proposed method upon the noise $\mathcal{N}$.

## A BACKGROUND

Inspired by outlier exposure (Hendrycks et al., 2018), most of the recently proposed methods generate pseudo anomalies by injecting random points into the context of time series. These methods typically rely on simple data augmentation strategies and fail to delve deeply into the prior knowledge underlying complex anomalies. Therefore, their generated anomalies are mostly confined to random anomalies or point anomalies. However, time series exhibit unique temporal dependencies and inherently encompass a variety of anomaly types. Such overly simplistic augmentation approaches fail to match the diverse nature of anomalies in real-world scenarios. They struggle to improve their detection performance. Unfortunately, they cannot cover the prevalent pattern anomalies in time series.

To deal with the two key types of anomalies, i.e., point anomaly and pattern anomaly, our proposed PAMA constructs a dual-branch augmentation framework. One branch in the framework is the trend perturbation branch. By sampling multi-dimensional linear trends, this branch disrupts the stable trend characteristics of normal time series. The other branch is the period perturbation branch. It can scale the dominant frequency components that influence time series variations in the frequency domain and break the seasonal regularity of normal data. Ultimately, by fusing the perturbed series output by the two branches, PAMA can generate pseudo anomalies that are similar to real-world scenarios.

However, noise rather than pseudo anomalies may be generated by applying random perturbations to the features of normal data. As illustrated in Fig. 5, it can be observed from Fig. 5(a) that three noise are misclassified as anomalies and one anomalous data as normal. Pseudo anomalies are introduced to assist in distinguishing between noise and anomalies, reducing the probability of misclassification as illustrated in Fig. 5(b).

Moreover, previous studies primarily focus on learning multi-scale features of time series through prototype. A single memory augmentation mechanism can only concentrate on capturing one specific type of pattern. Hence, it cannot simultaneously utilize normal and anomalous patterns. Through the dual-memory augmentation, the representations obtained by PAMA incorporate both normal and anomalous patterns. Fur-

thermore, temporal contrastive learning and instance contrastive learning are introduced in PAMA. As the result, the reconstruction error of normal data can be reduced.

## B SUPPLEMENT OF METHOD

In this section, some methods and formulas in PAMA are explained and supplemented in detail.

### B.1 PSEUDO-ANOMALY GENERATION

The PAG module for generating pseudo anomalies is an innovative method that employs a data augmentation strategy to generate pseudo anomalies by exploiting the prior knowledge of anomalies. Using PAG, PAMA can generate pseudo anomalies containing both trend and seasonal perturbations.

PAG performs random sampling on the original time series $\mathbf{X} \in \mathbb{R}^{T \times C}$. Its sampling position, sampling rate and sampling length are respectively set to be $p_s$, $r_s$ and $l_s = r_s \times T$ with $0 \le p_s \le T - l_s$. The sampled time series can be obtained as $\mathbf{X}_s \in \mathbb{R}^{l_s \times C}$. PAG generates a random trend perturbation $\mathbf{T}_p \in \mathbb{R}^{l_s \times C}$. For each channel $\mathbf{T}_p^i$, where $i \in \{1, \ldots, C\}$, the direction $d_i$ and amplitude $a_i$ are randomly selected. Let $\mathbf{T}_p^i(t)$ be the linear perturbation at the time step $t$, which is given by

$$\mathbf{T}_p^i(t) = t \frac{a_i}{l_s - 1} d_i. \tag{15}$$

The trend perturbation series $\mathbf{X}_{\text{tp}} \in \mathbb{R}^{l_s \times C}$ can thus be obtained by adding $\mathbf{T}_p$ and $\mathbf{X}_s$, that is

$$\mathbf{X}_{\text{tp}} = \mathbf{T}_p + \mathbf{X}_s. \tag{16}$$

Meanwhile, the original time series is converted into the frequency domain by FFT. We can thus obtain

$$\mathbf{X}(f) = \mathcal{F}(\mathbf{X}(t)) = \sum_{t=1}^{T} \mathbf{X}(t) e^{-j 2\pi \frac{f}{T} t}, \tag{17}$$

where $\mathcal{F}(\cdot)$ denotes the FFT function. Furthermore, PAG calculates the amplitude of each frequency component for all features in the frequency domain as follow.

$$\mathbf{A}(f) = |\mathbf{X}(f)| = \sqrt{\mathfrak{Re}\left(\mathbf{X}(f)\right)^2 + \mathfrak{Im}\left(\mathbf{X}(f)\right)^2}, \tag{18}$$

where $\mathfrak{Re}\left(\mathbf{X}(f)\right)$ and $\mathfrak{Im}\left(\mathbf{X}(f)\right)$ represent the real part and imaginary part of $\mathbf{X}(f)$, respectively.

PAG selects the frequency component $f_i^{\text{max}}$ with the largest amplitude on the $i$-th feature, that is

$$f_i^{\text{max}} = \arg\max_f \mathbf{A}(f, i). \tag{19}$$

Subsequently, PAG replaces the component in $\tilde{\mathbf{X}}(f)$ at frequency $f_i^{\text{max}}$ with the scaled component $\alpha$ while keeping the remaining components unchanged, that is

$$\tilde{\mathbf{X}}(f) = \begin{cases} \alpha \cdot \mathbf{X}(f) & \text{if } f = f_{\text{max}}^i \\ \mathbf{X}(f) & \text{otherwise} \end{cases} \tag{20}$$

$\tilde{\mathbf{X}}$ is thus converted back to the time domain by iFFT to obtain $\mathbf{X}_{\text{sp}}$, i.e.,

$$\mathbf{X}_{\text{sp}}(t) = \mathcal{F}^{-1}\left(\tilde{\mathbf{X}}(f)\right). \tag{21}$$

Finally, part of the final pseudo anomalies $\mathbf{X}_a$ with random position $p_g$ as the starting point and length $l_s$ is replaced by the trend perturbation series $\mathbf{X}_{\text{tp}}$, and the rest is obtained from the seasonal perturbation series $\mathbf{X}_{\text{sp}}$ of the corresponding time points. The process can be expressed as follows.

$$\mathbf{X}_a(i, j) = \begin{cases} \mathbf{X}_{\text{tp}}(i - p_g, j), & \text{if } p_g \le i \le p_g + l_s - 1; \\ \mathbf{X}_{\text{sp}}(i, j), & \text{Otherwise.} \end{cases} \tag{22}$$

## B.2 GL-ENCODER

PAMA adopts the global-local encoder (GL-Encoder) of UR-DMU. GL-Encoder uses Transformer to capture long-term trends and learn global dependencies, while utilizes convolutional neural network to capture short-term fluctuations and learn local dependencies.

The GL-Encoder inputs the time series $\mathbf{X}$ into the embedding layer to obtain its corresponding feature representations $\mathbf{Z} \in \mathbb{R}^{T \times d_{\text{model}}}$ by

$$\mathbf{Z} = \text{Embedding}(\mathbf{X}). \tag{23}$$

Assuming that GL-Encoder contains $L$ layers, the calculation process of the $l$-th layer can be expressed as

$$\begin{aligned}
\mathbf{A}^l &= \text{LayerNorm}\big(\text{GL-Encoder}(\mathbf{Z}^{l-1}) + \mathbf{Z}^{l-1}\big), \\
\mathbf{Z}^l &= \text{LayerNorm}\big(\text{Feed-Forward}(\mathbf{A}^l) + \mathbf{A}^l\big),
\end{aligned} \tag{24}$$

where $\mathbf{Z}^l, \mathbf{A}^l \in \mathbb{R}^{T \times d_{\text{model}}}$, $l \in \{1, \ldots, L\}$.

The GL-Encoder at the $h$-th head of the $l$-th layer is denoted by

$$\begin{aligned}
[\mathbf{Q}_h, \mathbf{K}_h, \mathbf{V}_h] &= \left[\mathbf{Z}^{l-1}\mathbf{W}_Q^{(l,h)}, \mathbf{Z}^{l-1}\mathbf{W}_K^{(l,h)}, \mathbf{Z}^{l-1}\mathbf{W}_V^{(l,h)}\right], \\
\mathbf{T}_h &= \mathbf{Z}^{l-1}\mathbf{W}_T^{(l,h)}, \\
\mathbf{D} &= -\frac{|i-j|}{e^\tau},
\end{aligned} \tag{25}$$

where $\mathbf{W}_Q^{(l,h)}, \mathbf{W}_K^{(l,h)}, \mathbf{W}_V^{(l,h)}, \mathbf{W}_T^{(l,h)} \in \mathbb{R}^{d_{\text{model}} \times \frac{d_{\text{model}}}{H}}$, $H$ represents the number of attention heads, and $\tau$ is the sensitivity factor to balance the l0cal distance. GL-Encoder computes the global and local attention through the obtained $\mathbf{Q}_h, \mathbf{K}_h, \mathbf{V}_h, \mathbf{T}_h \in \mathbb{R}^{T \times \frac{d_{\text{model}}}{H}}$ and $\mathbf{D} \in \mathbb{R}^{T \times T}$, where $\mathbf{D}(i,j)$ represents the relative time distance between time steps $i$ and $j$. Thus, GL-Encoder obtain the global representations $\mathbf{Z}_G^{(l,h)}$ and the local representations $\mathbf{Z}_L^{(l,h)}$. GL-Encoder combines the representations of multiple heads and $\mathbf{Z}_G^l$ and $\mathbf{Z}_L^l$ are concatenated to obtain the representations $\mathbf{Z}^l$ through an MLP projection layer. The detailed process is given by

$$\begin{aligned}
\mathbf{Z}_G^{(l,h)} &= \text{Softmax}\left(\frac{\mathbf{Q}_h\mathbf{K}_h^{\mathrm{T}}}{\sqrt{d_{\text{model}}}}\right)\mathbf{V}_h, \\
\mathbf{Z}_L^{(l,h)} &= \text{Softmax}\left(\mathbf{D}\mathbf{T}_h\right), \\
\mathbf{Z}_G^l &= \text{Concat}\left(\mathbf{Z}_G^{(l,1)}, \mathbf{Z}_G^{(l,2)}, \ldots, \mathbf{Z}_G^{(l,H)}\right), \\
\mathbf{Z}_L^l &= \text{Concat}\left(\mathbf{Z}_L^{(l,1)}, \mathbf{Z}_L^{(l,2)}, \ldots, \mathbf{Z}_L^{(l,H)}\right), \\
\mathbf{Z}^l &= \text{Projection}\left(\text{Concat}(\mathbf{Z}_G^l, \mathbf{Z}_L^l)\right).
\end{aligned} \tag{26}$$

PAMA can obtain $\mathbf{Z}$ containing both global and local information by using GL-Encoder.

## B.3 MEMORY MODULE GENERATION

The dual-memory module generation process follows the MEMTO design and uses a two-stage update method. Let MMG and Update represent the memory module generation function and feature representation update function, respectively. This yields the normal memory module $\mathbf{M}_n$, which records normal prototypes, and the pseudo-anomalous memory module $\mathbf{M}_a$, which records pseudo-anomalous prototypes, that is

$$\begin{aligned}
\mathbf{M}_n &= \text{MMG}(\mathbf{Z}_n), \\
\mathbf{M}_a &= \text{MMG}(\mathbf{Z}_a).
\end{aligned} \tag{27}$$

DMA uses the dual memory module to update the normal feature representation $\mathbf{Z}_n$ and the pseudo-anomalous feature representation $\mathbf{Z}_a$ separately, the Updateprocedure is represented as

$$
\begin{aligned}
\mathbf{Z}_n &= \text{Update}(\mathbf{Z}_n, \mathbf{M}_n), \\
\mathbf{Z}_a &= \text{Update}(\mathbf{Z}_a, \mathbf{M}_a).
\end{aligned}
\tag{28}
$$

MMG is the gated memory update stage in MEMTO, and Update is the query update stage. The dual-memory module of PAMA also adopts the two-stage training in MEMTO without modifying it.

### B.4 UNCERTAINTY LEARNING

PAMA modifies the conventional uncertainty learning (UL) to effectively learn data uncertainty. Through $\text{Encoder}_\mu$ and $\text{Encoder}_{var}$, we can obtain the mean vector $\boldsymbol{\mu}_n^d$ and variance $(\sigma_n^d)^2$ for each variate of $\bar{\mathbf{Z}}_{nn}$. Moreover, $\boldsymbol{\mu}_a^d$ can also be achieved for each variate of $\bar{\mathbf{Z}}_{aa}$. Hence, we can get

$$
\boldsymbol{\mu}_n^d = \text{Encoder}_\mu\left(\bar{\mathbf{Z}}_{nn}\right), \quad \boldsymbol{\mu}_a^d = \text{Encoder}_\mu\left(\bar{\mathbf{Z}}_{aa}\right), \quad (\sigma_n^d)^2 = \text{Encoder}_{var}\left(\bar{\mathbf{Z}}_{nn}\right).
\tag{29}
$$

UL constraints the $d$-th variate of $\bar{\mathbf{Z}}_{nn}$, i.e., $\bar{\mathbf{z}}_n^d$ to obey the normal distribution $\mathcal{N}\left(\boldsymbol{\mu}_n^d, \left(\sigma_n^d\right)^2 \mathbf{I}\right)$. Note that $\bar{\mathbf{z}}_n^d$ is generated by the reparameterization technique to ensure its differentiability. Thus,

$$
\bar{\mathbf{z}}_n^d = \boldsymbol{\mu}_n^d + \sigma_n^d \boldsymbol{\varepsilon}, \quad \boldsymbol{\varepsilon} \sim \mathcal{N}(\mathbf{0}, \mathbf{I}).
\tag{30}
$$

To suppress instability caused by excessive $\sigma_n^d$, $\mathcal{L}_{\text{k1}}$ is used as the regularization loss, i.e.,

$$
\mathcal{L}_{\text{k1}} = \text{KL}\left(\mathcal{N}(\bar{\mathbf{z}}_n | \boldsymbol{\mu}_n, \sigma_n) \parallel \mathcal{N}(0, 1)\right) = -\frac{1}{2 d_{\text{model}}} \sum_{d=1}^{d_{\text{model}}} \left(1 + \log(\sigma_n^d)^2 - \|\boldsymbol{\mu}_n^d\|^2 - (\sigma_n^d)^2\right).
\tag{31}
$$

To simultaneously maximize the distance between $\bar{\mathbf{z}}_n^d$ and $\bar{\mathbf{z}}_a^d$ while minimize the distance between $\bar{\mathbf{z}}_n^d$ and its corresponding cluster center, the following distance loss is adopted.

$$
\mathcal{L}_{\text{dis}} = \frac{1}{d_{\text{model}}} \sum_{d=1}^{d_{\text{model}}} \left[ \max\left(0, d_z - \left\|\boldsymbol{\mu}_a^d - \bar{\mathbf{z}}_n^d\right\|\right) + \left\|\boldsymbol{\mu}_n^d - \bar{\mathbf{z}}_n^d\right\| \right],
\tag{32}
$$

where $d_z$ is a hyperparameter controlling the feature distance.

Let the normal representations after UL be $\bar{\mathbf{Z}}_n$. $\bar{\mathbf{Z}}_n$ and $\mathbf{Z}_n$ are concatenated and then inputted into the decoder composed of linear layers. The reconstructed time series $\hat{\mathbf{X}}$ can be ultimately obtained as follows

$$
\hat{\mathbf{X}} = \text{Decoder}\left(\text{Concat}(\bar{\mathbf{Z}}_n, \mathbf{Z}_n)\right).
\tag{33}
$$

### B.5 ENTROPY LOSS

PAMA uses the entropy loss $\mathcal{L}_{\text{entr}}$ in MEMTO as the auxiliary loss for sparsification of $\mathbf{W}$ in the dual-memory module. Let $\mathbf{W}(t, i)$ denote the similarity between the representation at time step $t$ and the $i$-th prototype. Minimizing $\mathcal{L}_{\text{entr}}$ can ensure retrieving only a restricted number of closely relevant normal and anomalous prototypes in dual-memory module. Since PAMA adopts dual-memory module, it needs to sparsify $\mathbf{W}_n$ and $\mathbf{W}_a$. The entropy loss $\mathcal{L}_{\text{entr}}$ is expressed as

$$
\begin{aligned}
\mathcal{L}_{\text{entr}} = \sum_{t=1}^{T} \sum_{i=1}^{N} &-\mathbf{W}_n(t, i) \log\left(\mathbf{W}_n(t, i)\right) \\
&- \mathbf{W}_a(t, i) \log\left(\mathbf{W}_a(t, i)\right).
\end{aligned}
\tag{34}
$$

### B.6 ALGORITHM DESCRIPTION

The training and testing procedures of PAMA are summarized in Algorithm 1 and Algorithm 2, respectively.

---

**Algorithm 1** Training Procedure of PAMA

---

**Require:** Training set $\mathbf{X}$, Length of sliding window $T$, Batch size $B$, Learning rate $\eta$, Maximum training iterations $E$.
**Ensure:** Sets of the optimal network parameters $\boldsymbol{\theta}^*$.
 1: Initialize parameters $\boldsymbol{\theta}$.
 2: **while** $i \leq E$ **do**
 3:     $\mathbf{X}_a \leftarrow \text{PAG}(\mathbf{X}), \mathbf{X}_n \leftarrow \mathbf{X}$.
 4:     $\mathbf{Z}_p \leftarrow \text{GL-Encoder}(\mathbf{X}_p), p \in \{n, a\}$.
 5:     $\mathbf{M}_q \leftarrow \text{DMM}(\mathbf{Z}_q), q \in \{n, a\}$.
 6:     $\bar{\mathbf{Z}}_{pq} \leftarrow \text{DMA}(\mathbf{Z}_p, \mathbf{M}_q)$.
 7:     Compute $\mathcal{L}_{\text{temp}}$ by Eq. (7).
 8:     Compute $\mathcal{L}_{\text{inst}}$ by Eq. (8).
 9:     $\mathcal{L}_{\text{con}} \leftarrow \frac{1}{TB} \sum_{t=1}^{T} \sum_{i=1}^{B} (\mathcal{L}_{\text{temp}}^{(i,t)} + \mathcal{L}_{\text{inst}}^{(i,t)})$.
10:     Compute $\mathcal{L}_{\text{kl}}$ by Eq. (31) and compute $\mathcal{L}_{\text{dis}}$ by Eq. (32).
11:     $\hat{\mathbf{X}} \leftarrow \text{Decoder}\left(\text{Concat}(\bar{\mathbf{Z}}_n, \mathbf{Z}_n)\right)$.
12:     Compute $\mathcal{L}_{\text{rec}}$ by Eq. (11).
13:     Compute $\mathcal{L}$ by Eq. (12).
14:     Update parameters $\boldsymbol{\theta} \leftarrow \boldsymbol{\theta} - \eta \cdot \nabla_{\boldsymbol{\theta}} \mathcal{L}$.
15:     $i = i + 1$.
16: **end while**
17: **return** $\boldsymbol{\theta}^* = \boldsymbol{\theta}$.

---

---

**Algorithm 2** Testing procedure of PAMA

---

**Require:** Test set $\mathbf{X}_{\text{test}} = \{\mathbf{x}_t\}_{t=1}^{L}$, Abnormal rate $\delta$, Sets of the optimal network parameters $\boldsymbol{\theta}^*$.
**Ensure:** Anomaly scores $\{s_t\}_{t=1}^{L}$, Class labels $\{y_t\}_{t=1}^{T}$.
 1: Reconstruct all time steps in $\mathbf{X}_{\text{test}}$ by PAMA.
 2: Compute $\{s_t\}_{t=1}^{L}$ by Eq. (13).
 3: Sort $\{s_t\}_{t=1}^{L}$ and determine $\xi$ according to $\delta$.
 4: **for** $t = 1$ to $L$ **do**
 5:     Compute $y_t$ by Eq. (14).
 6: **end for**
 7: **return** $\{s_t\}_{t=1}^{L}$ and $\{y_t\}_{t=1}^{L}$.

---

## C SUPPLEMENT OF EXPERIMENTAL SETUP

In this section, we supplement some experimental setups.

### C.1 DATASETS

In the experiments, seven commonly used benchmark datasets for MTSAD are utilized. The detailed information of the seven datasets is summarized in Table 3. Moreover, the brief description of them is as follows.

Table 3: Details of seven benchmark datasets used for time series anomaly detection.

| Datasets | $N_{\text{D}}$ | $N_{\text{tr}}$ | $N_{ts}$ | $r$ |
|---|---|---|---|---|
| MSL | 55 | 58317 | 73729 | 10.50 |
| SMAP | 25 | 135183 | 427617 | 12.80 |
| PSM | 25 | 132481 | 87841 | 27.80 |
| SWaT | 51 | 475200 | 449919 | 12.14 |
| SMD | 51 | 708405 | 708420 | 4.20 |
| NIPS-TS-SWAN | 38 | 60000 | 60000 | 32.60 |
| NIPS-TS-GECCO | 9 | 69260 | 69261 | 1.10 |

Note: $N_{\text{D}}$–Number of dimensions; $N_{\text{tr}}$–Number of training data; $N_{\text{ts}}$–Number of test data; $r$–Anomaly rate (%).

- MSL (Mars Science Laboratory) (Hundman et al., 2018) is an important resource for scientific research on Mars, providing a large number of high-precision data on the surface, climate, geology and life potential of Mars.

- SMAP (Soil Moisture Active Passive satellite) (Hundman et al., 2018) is a remote sensing dataset about soil moisture developed by NASA, which aims to provide detailed soil moisture information for climate research, agricultural monitoring, water resources management and other fields .

- PSM(Pooled Server Metrics) (Abdulaal et al., 2021) is a related dataset used to monitor, analyze, and optimize the performance of computer servers or data centers. This dataset records various performance metrics of the server, usually including but not limited to processor (CPU) usage, memory usage, disk IO, network traffic, load balance, error rate, and other critical performance parameters.

- SWaT (Secure Water Treatment) (Goh et al., 2016) is a public dataset for security evaluation of industrial control systems in the context of water treatment facilities. This dataset is designed to support researchers and engineers in identifying, monitoring, and preventing security vulnerabilities in water treatment facilities, especially against potential cyber attacks or system failures.

- SMD (Server Machine Dataset) (Su et al., 2019) is a dataset used for machine learning and anomaly detection tasks, mainly derived from machine performance monitoring data of servers and data centers. This dataset is designed to help researchers and engineers monitor and predict performance issues of servers in high load and stress environments, especially in data centers and cloud computing environments. Through this dataset, it is possible to study the running state of the server machine, perform performance optimization, anomaly detection, and even provide a basis for failure prevention.

- NIPS-TS-SWAN (Angryk et al., 2020) is a multivariate time series extracted from the solar photosphere vector magnetic map, while NIPS-TS-GECCO (Ribeiro & Reynoso-Meza, 2018) is a dataset for drinking water quality detection.

## C.2 EVALUATION CRITERIA

In the experiments, three commonly used metrics, namely, Precision (Pre), Recall (Rec) and F1-score (F1) are used to evaluate and compare the proposed PAMA with the baseline method. Their expressions are as

follow.

$$\text{Pre} = \frac{\text{TP}}{\text{TP} + \text{FP}},$$
$$\text{Rec} = \frac{\text{TP}}{\text{TP} + \text{FN}}, \quad (35)$$
$$\text{F1} = \frac{2 \times \text{Pre} \times \text{Rec}}{\text{Pre} + \text{Rec}},$$

Where TP, FP and FN are respectively the number of true positive, false positive and false negative samples. It should be noted that PAMA adopts the point adjustment strategy commonly used in MTSAD. Since F1-score and point adjustment strategies have been subject to some controversy in recent years(Kim et al., 2021), other evaluation metrics are introduced and compared with baseline methods. Such a practice aims to further verify the effectiveness and detection performance of PAMA. They include Aff-p, Aff-r, Range-AUC-ROC, Range-AUC-PR, V-ROC, V-PR, AUC-ROC and AUC-PR.

### C.3 SUPPLEMENT OF IMPLEMENTATION DETAILS

To ensure the reproducibility of the experiments, some parameter settings of PAMA are supplemented. The sliding window length $T$ and moving step size $s$ are both taken as 100, and the latent space dimension $d_{\text{model}}$ is assigned with 512. The encoder possesses three attention network layers and eight attention heads, the decoder contains only one linear layer, and the GELU activation function is used. The Adam optimization algorithm with an initial learning rate of $10^{-4}$ in the first stage and $5 \times 10^{-5}$ in the second stage is adopted. The batch size $B$ is set to 32, and the maximum number of training epochs is taken as 100. We use ten proto-types for each module in dual-memory module of PAMA. Some hyperparameters of PAMA are determined by grid search, while others are chosen by empirical observations. In addition, PAMA is implemented based on PyTorch, and all experiments are conducted on a single NVIDIA 4090 24GB GPU.

## D SUPPLEMENT OF EXPERIMENTS

### D.1 COMPARISON EXPERIMENT

To deeply verify the detection performance of PAMA, we specially carry out experiments on the more challenging NIPS-TS-GECCO and NIPS-TS-SWAN datasets. Compared with the previous five datasets, these two datasets cover a richer variety of anomaly types, which can test the detection ability of the model more comprehensively. The testing results of different methods are summarized in Table 4, which provides strong support for evaluating the detection performance of PAMA in complex scenarios. Furthermore, to comprehensively validate the efficiency of PAMA, it is compared with the other baseline methods on the seven datasets, further validating the superiority of PAMA. It is worth noting that the F1 scores of most baseline methods are significantly lower than those of PAMA. Therefore, we only select baseline methods from recent years for evaluation, including Anomaly Transformer, DCdetector, U-Transformer, and MEMTO, and compare them across six evaluation metrics. The results are included in Table 5. By analyzing the results in Tables 4-5, it can be observed that PAMA demonstrates higher accuracy and better generalization performance when handling different anomaly rates and various anomaly datasets.

### D.2 ABLATION EXPERIMENTS

In this subsection, we check the necessity of each component in PAMA by ablation experiments. Table 6 shows the effect of different manners of pseudo-anomaly generation on the testing results of PAMA upon the four benchmark datasets. According to the results in Table 6, it can be seen that in most cases, the usage of trend and seasonal perturbations can improve the detection performance of PAMA. Moreover, introducing

Table 4: Testing results of PAMA and the five baseline methods on the NIPS-TS datasets.(%).

| Datasets | NIPS-TS-GECCO | | | NIPS-TS-SWAN | | |
|---|---|---|---|---|---|---|
| Models | Pre | Rec | F1 | Pre | Rec | F1 |
| Anomaly Trans | 30.40 | 47.80 | 37.20 | 96.60 | 59.30 | 73.50 |
| MEMTO | 40.51 | 67.53 | 50.64 | **98.18** | 60.45 | 74.82 |
| U-Transformer | 2.84 | 77.95 | 5.47 | 98.07 | 61.77 | 75.80 |
| DCdetector | 37.40 | 56.60 | 45.10 | 97.00 | 59.30 | 73.60 |
| H-PAD | 50.64 | 55.61 | 60.78 | 97.64 | 62.61 | 76.28 |
| PAMA | **65.65** | **100.00** | **79.26** | 94.59 | **64.07** | **76.40** |

pseudo anomalies can significantly improve the performance of PAMA, indicating that our research on pseudo anomalies is effective. Especially, by utilizing pseudo anomalies, PAMA can capture anomalous patterns by using pseudo-anomalous memory module.

The complete ablation study for pseudo-anomaly contrastive learning is shown in Table 7 and the complete ablation experiment for each component of PAMA is included in Table 8.

### D.3 PARAMETER SENSITIVITY EXPERIMENTS

To explore the impact of different weight values for their corresponding loss functions on the detection performance of PAMA, we adjust the weight values and keep the other hyperparameters fixed. The testing results are illustrated in Fig. 6. From Fig. 6 it can easily be found that PAMA achieves higher stability on PSM under different hyperparameter values. Except for $\lambda_3$, PAMA achieves stable F1-scores on SMAP under different hyperparameter values. Moreover, PAMA is sensitive to the hyperparameter values on MSL because the F1-scores exhibit significant fluctuations as the hyperparameter values change.

Furthermore, we conduct parameter sensitivity analysis on the dimension of latent space $d_{model}$, temperature parameters $\tau$, and the number of prototypes $n$. The outcomes are summarized in Fig. 7. One can observe from Fig. 7(a) that when $d_{model} = 512$, PAMA achieves the best performance on the three datasets, and when $d_{model} = 1024$, the performance is suboptimal. Hence, PAMA can efficiently distinguish between normal and anomalous representations in high-dimensional space. It can be found from Fig. 7(b) that the performance of PAMA gradually decreases as the value of $\tau$ increases. Moeover, we can find from Fig. 7(c) that as the number of prototypes increases, the performance of PAMA slightly decreases on SMAP and PSM, but greatly fluctuates on MSL. This indicates that PAMA may learn similar prototypes on MSL and cannot take advantage of the dual-memory module to learn both normal and anomalous patterns.

### D.4 MODEL ANALYSIS

Further experiments are conducted to compare PAMA with the baseline model MEMTO. The comparison is made from four aspects, such as the ability to deal with irregular and sparse time series, robustness to different ratios of noise, robustness to different lengths of time series, and robustness to different types of noise. Correspondingly, we carry out numerical experiments to investigate how the F1-score changes with different values of the masking ratio $r_{mask}$, the noise intensity $r_{noise}$, the window size $T$ and different types of noise. The results are summarized in Fig. 8.

Overall, PAMA exhibits significant superiority over MEMTO in anomaly detecting performance. First, sparse times series are generated by randomly masking the series with ratio $r_{mask}$. As shown in Fig. 8(a), PAMA exhibits better stability than MEMTO in terms of F1-score. Even when the series has high sparsity $r_{mask} = 50\%$, F1-score of PAMA still keep in a high level. Second, noise robustness is studied by introducing

Table 5: Testing results of PAMA and the four baseline methods on the NIPS-TS datasets (%), where R-A-R and R-A-P respectively denote Range-AUC-ROC and Range-AUC-PR.

| Datasets | Models | Aff-p | Aff-r | R-A-R | R-A-P | V-ROC | V-PR |
|---|---|---|---|---|---|---|---|
| MSL | Anomaly Trans | 51.15 | 96.35 | 90.34 | 88.15 | 89.48 | 87.41 |
| | DCdetector | 52.74 | **97.22** | **91.86** | **89.54** | 90.99 | 88.79 |
| | U-Transformer | **56.46** | 91.19 | 82.49 | 80.72 | 82.25 | 80.47 |
| | MEMTO | 51.43 | 96.00 | 90.36 | 88.19 | 89.40 | 87.38 |
| | H-PAD | 55.98 | 96.25 | 91.34 | 88.66 | 89.94 | 87.91 |
| | PAMA | 51.80 | 96.33 | 91.36 | 89.15 | **91.21** | **89.04** |
| SMAP | Anomaly Trans | 50.41 | 97.48 | 96.08 | 93.99 | 95.46 | 93.46 |
| | DCdetector | 51.36 | 98.40 | 93.71 | 92.22 | 93.50 | 92.02 |
| | U-Transformer | 41.50 | 66.73 | 62.58 | 64.17 | 62.79 | 64.35 |
| | MEMTO | **52.89** | 98.87 | 96.02 | 93.83 | 95.33 | 93.23 |
| | H-PAD | 52.46 | 98.91 | 96.83 | 94.13 | 95.86 | 93.30 |
| | PAMA | 51.36 | **98.96** | **97.00** | **94.68** | **96.74** | **94.47** |
| PSM | Anomaly Trans | 54.94 | 82.80 | 91.68 | 93.28 | 90.04 | 92.08 |
| | DCdetector | 54.85 | 82.16 | 88.95 | 91.03 | 85.41 | 88.39 |
| | U-Transformer | **86.95** | 49.09 | 73.62 | 81.45 | 72.61 | 80.7 |
| | MEMTO | 56.86 | 84.79 | 91.80 | 93.43 | 89.89 | 92.04 |
| | H-PAD | 60.29 | 84.85 | 92.91 | 92.42 | 90.65 | 91.70 |
| | PAMA | 56.18 | **85.25** | **94.20** | **95.09** | 91.78 | **93.36** |
| SWaT | Anomaly Trans | 53.25 | **97.99** | **97.88** | 93.40 | **97.91** | 93.43 |
| | DCdetector | 56.21 | 96.27 | 96.43 | **93.95** | 95.80 | 93.40 |
| | U-Transformer | **99.81** | 2.83 | 51.15 | 57.18 | 51.13 | 57.16 |
| | MEMTO | 59.04 | 93.41 | 89.04 | 87.40 | 89.19 | 87.53 |
| | H-PAD | 60.40 | 87.45 | 88.82 | 85.18 | 86.31 | 84.51 |
| | PAMA | 57.24 | 88.76 | 87.40 | 86.04 | 87.57 | 86.18 |
| SMD | Anomaly Trans | 59.50 | 89.47 | 74.90 | 71.35 | 75.18 | 71.64 |
| | DCdetector | 50.75 | 90.15 | 72.04 | 67.54 | 70.04 | 65.63 |
| | U-Transformer | **79.71** | 74.54 | 70.96 | 58.01 | 71.77 | 58.79 |
| | MEMTO | 61.34 | 94.81 | 80.35 | 76.57 | 81.01 | 77.23 |
| | H-PAD | 61.91 | 93.22 | 81.02 | **78.86** | 82.02 | 79.15 |
| | PAMA | 58.20 | **97.74** | **83.30** | 78.43 | **84.23** | **79.38** |
| NIPS-TS-GECCO | Anomaly Trans | 54.98 | 85.46 | 60.05 | 27.66 | 58.41 | 26.02 |
| | DCdetector | 51.29 | 88.49 | 62.32 | 33.19 | 61.89 | 32.84 |
| | U-Transformer | **72.98** | 70.38 | 50.14 | 17.05 | 51.77 | 18.68 |
| | MEMTO | 56.5 | 90.32 | 64.46 | 36.63 | 64.33 | 36.49 |
| | H-PAD | 57.87 | 93.21 | 71.12 | **70.16** | 69.03 | 48.64 |
| | PAMA | 58.81 | **96.69** | **71.93** | 48.15 | **75.01** | **51.24** |
| NIPS-TS-SWAN | Anomaly Trans | 52.25 | 4.41 | 88.07 | 94.84 | 86.20 | 93.63 |
| | DCdetector | 56.16 | 4.30 | 88.07 | 94.89 | 86.19 | 93.67 |
| | U-Transformer | **93.32** | 17.93 | 88.01 | **94.9** | **86.34** | **93.86** |
| | MEMTO | 78.08 | 5.59 | 88.09 | 94.85 | 86.27 | 93.69 |
| | H-PAD | 59.17 | 73.15 | 84.82 | 79.65 | 85.14 | 84.47 |
| | PAMA | 71.68 | **75.54** | **88.12** | 94.85 | 86.30 | 93.68 |
| Average | Anomaly Trans | 53.79 | 79.14 | 85.57 | 80.38 | 84.67 | 79.67 |
| | DCdetector | 53.33 | 79.57 | 84.77 | 80.34 | 83.40 | 79.25 |
| | U-Transformer | **75.81** | 53.24 | 68.42 | 64.78 | 68.38 | 64.86 |
| | MEMTO | 59.45 | 80.54 | 85.73 | 81.56 | 85.06 | 81.08 |
| | H-PAD | 58.29 | 89.57 | 86.69 | **84.15** | 85.56 | 81.38 |
| | PAMA | 57.90 | **91.32** | **87.62** | 83.77 | **87.55** | **83.90** |

Table 6: Testing results of PAMA using different manners of pseudo-anomaly generation on the four benchmark datasets (%).

| PAG | | MSL | | | SMAP | | | PSM | | | SWaT | | |
|---|---|---|---|---|---|---|---|---|---|---|---|---|---|
| Trend | Seasonal | Pre | Rec | F1 | Pre | Rec | F1 | Pre | Rec | F1 | Pre | Rec | F1 |
| ✗ | ✗ | 92.13 | 98.39 | 95.16 | 94.16 | 99.29 | 96.66 | 98.51 | 99.46 | 99.02 | 87.77 | 97.54 | 94.40 |
| ✓ | ✗ | 92.24 | 99.06 | 95.53 | 95.35 | 99.21 | 97.24 | 98.59 | 99.41 | 99.00 | 97.62 | 88.81 | 95.01 |
| ✗ | ✓ | 92.16 | 95.79 | 95.36 | 95.25 | 98.42 | 96.81 | 98.88 | 99.42 | **99.15** | 90.15 | 97.54 | 95.10 |
| ✓ | ✓ | 94.34 | 97.93 | **96.10** | 95.95 | 98.95 | **97.42** | 98.96 | 99.35 | 99.11 | 92.05 | 99.29 | **95.53** |

Note: Trend–Trend Perturbation; Seasonal–Seasonal Perturbation.

Table 7: Testing results of different contrastive learning on the four benchmark datasets (%).

| PACL | | MSL | | | SMAP | | | PSM | | | SWaT | | |
|---|---|---|---|---|---|---|---|---|---|---|---|---|---|
| Temporal | Instance | Pre | Rec | F1 | Pre | Rec | F1 | Pre | Rec | F1 | Pre | Rec | F1 |
| ✗ | ✗ | 92.54 | 93.57 | 95.05 | 94.70 | 98.93 | 96.47 | 98.61 | 99.35 | 98.98 | 95.62 | 92.81 | 94.01 |
| ✓ | ✗ | 90.69 | 98.83 | 95.28 | 95.82 | 97.42 | 96.9 | 98.85 | 99.32 | 99.05 | 89.88 | 97.54 | 94.75 |
| ✗ | ✓ | 94.84 | 96.42 | 95.35 | 95.30 | 98.55 | 96.61 | 98.86 | 99.35 | 99.08 | 94.12 | 89.12 | 94.55 |
| ✓ | ✓ | 94.34 | 97.93 | **96.10** | 95.95 | 98.95 | **97.42** | 98.86 | 99.35 | **99.11** | 92.05 | 99.29 | **95.53** |

Note: Temporal–Temporal Contrast; Instance–Instance Contrast.

Table 8: Test results of different components of the model on four benchmark datasets.(%).

| Model | | | MSL | | | SMAP | | | PSM | | | SWaT | | |
|---|---|---|---|---|---|---|---|---|---|---|---|---|---|---|
| DMA | PACL | UL | Pre | Rec | F1 | Pre | Rec | F1 | Pre | Rec | F1 | Pre | Rec | F1 |
| ✗ | ✗ | ✗ | 92.24 | 99.06 | 95.05 | 93.58 | 99.27 | 96.30 | 97.46 | 98.38 | 98.52 | 98.07 | 97.54 | 94.26 |
| ✓ | ✗ | ✗ | 92.23 | 99.33 | 95.15 | 97.51 | 99.50 | 96.35 | 98.86 | 99.35 | 98.61 | 96.07 | 98.07 | 94.3 |
| ✓ | ✗ | ✓ | 92.54 | 93.57 | 95.55 | 94.70 | 98.93 | 96.87 | 98.61 | 99.35 | 98.98 | 95.62 | 92.81 | 94.91 |
| ✓ | ✓ | ✗ | 95.32 | 90.23 | 95.81 | 95.33 | 99.31 | 96.78 | 96.8 | 98.87 | 99.02 | 97.62 | 88.81 | 95.1 |
| ✓ | ✓ | ✓ | 94.34 | 97.93 | **96.10** | 95.95 | 98.95 | **97.42** | 98.86 | 99.35 | **99.11** | 92.05 | 99.29 | **95.53** |

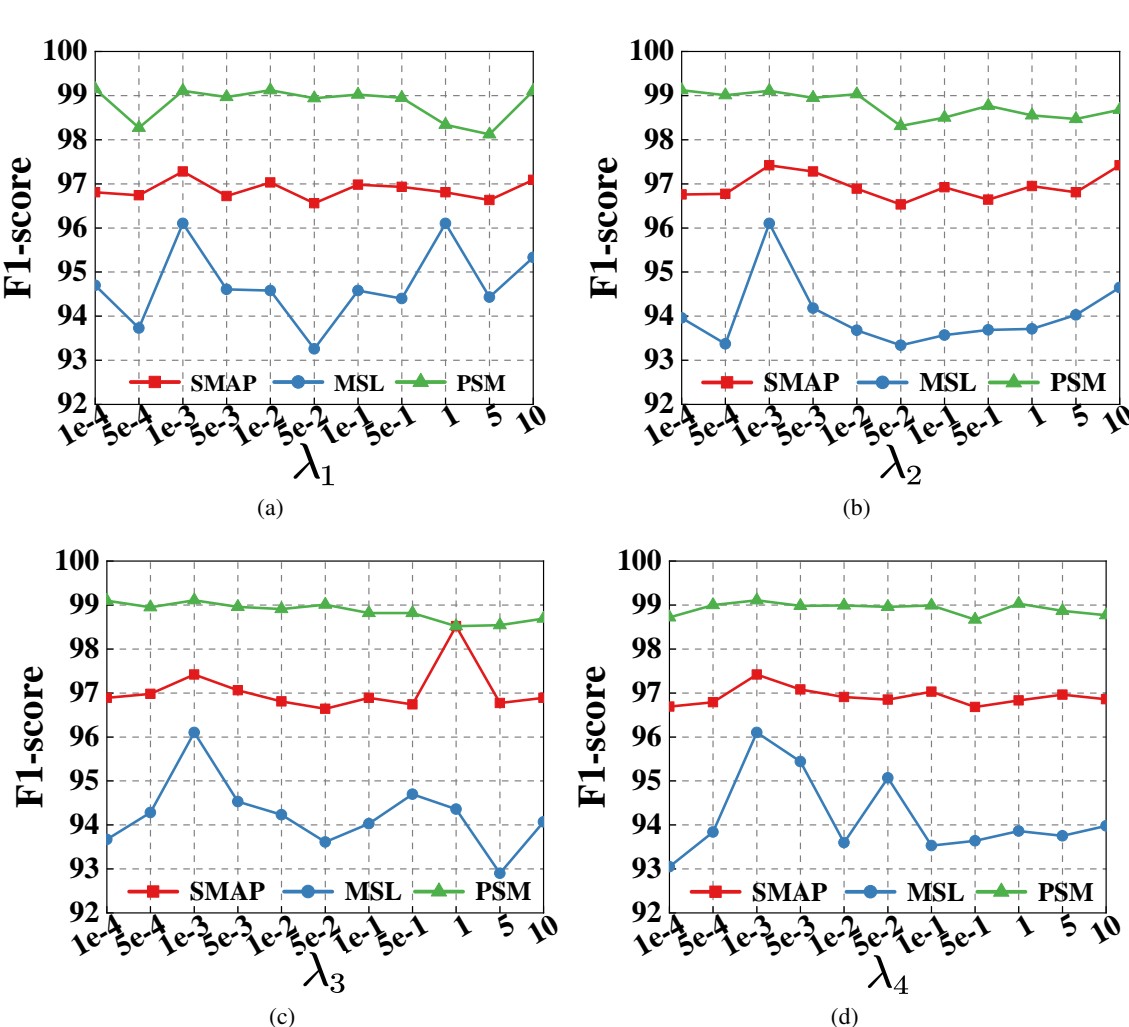

Figure 6: Influence of different values of weights on the detection performance of PAMA upon the three benchmark datasets. (a) The influence of different values of $\lambda_1$; (b) The influence of different values of $\lambda_2$; (c) The influence of different values of $\lambda_3$; (d) The influence of different values of $\lambda_4$.

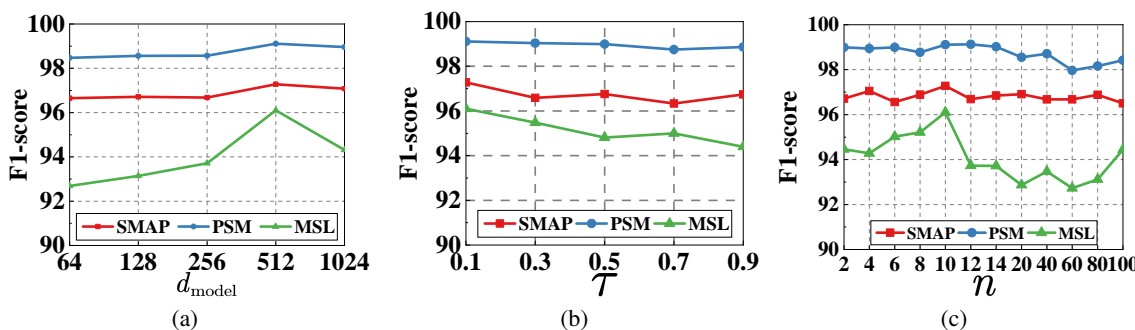

(a)          (b)          (c)

Figure 7: Influence of different vaules of hyperparameters on the performance of PAMA upon the three benchmark datasets. (a) The effect of different values of the latent space dimension $d_{\mathrm{model}}$; (b) The effect of different values of the temperature parameter $\tau$; (c) The effect of different number of prototype $n$.

Gaussian noise with the mean value of 0 and the variance of 0.5 into the training set. It can be observed from Fig. 8(b) that F1 scores of both models decrease as the noise rate $r_{\mathrm{noise}}$ increases. However, F1-score of PAMA show significantly slower shrinkage, verifying the effectiveness of introducing pseudo anomalies into PAMA. Effectively learning anomalous patterns enables PAMA to demonstrate excellent robustness under noise interference, consequently significantly mitigating the negative impact of noise. Third, Fig. 8(c) shows that PAMA maintains a higher F1-score with different window lengths and exhibits greater stability, indicating its superior temporal information extraction capability than MEMTO. Finally, noise-resistant experiments are conducted by introducing five types of noise. The experiments are conducted under $r_{\mathrm{noise}} = 0.2$. The average performance on the five benchmark datasets is shown in Fig. 8(d). PAMA maintains a high F1-score under all the five types of noise, demonstrating excellent noise resistance and robustness. Especially, PAMA can not only withstand a single type of noise, but also adapt to multiple types of noise with minimal performance fluctuations. This indicates that the contrastive learning method based on pseudo anomalies can effectively mitigate noise interference.

To observe the influence of the point adjustment strategy on the generalization performance of PAMA, the results with and without point adjustment across the four methods are summarized in Table 9. Compared to the other three methods, PAMA achieves higher average performance regardless of whether point adjustment is applied. Moreover, PAMA gets the smallest inflation gap value.

Table 9: Testing results of PAMA and the baseline methods on the five datasets (%), where F1* denotes F1-score without using point adjustment and Inf. Gap denotes Inflation Gap.

| Models | MSL | | SMAP | | PSM | | SWaT | | SMD | | Average | | |
|---|---|---|---|---|---|---|---|---|---|---|---|---|---|
| | F1 | F1* | F1 | F1* | F1 | F1* | F1 | F1* | F1 | F1* | F1 | F1* | Inf. Gap |
| MEMTO | 94.36 | 22.70 | 96.44 | 27.23 | 98.18 | 52.15 | 93.25 | 29.01 | 94.00 | 51.69 | 95.24 | 36.56 | 58.68 |
| H-PAD | 95.45 | 50.11 | 97.21 | 53.25 | **99.12** | 56.11 | **95.09** | **76.07** | 95.53 | **59.79** | 96.46 | 59.06 | 37.04 |
| CARLA | 96.07 | 52.27 | 96.97 | 52.92 | 99.09 | **60.37** | 95.01 | 72.09 | **95.66** | 51.14 | 96.56 | 57.76 | 38.80 |
| PAMA | **96.10** | **52.35** | **97.42** | **54.15** | 99.11 | 59.35 | 94.84 | 75.19 | 95.53 | 58.64 | **96.60** | **59.93** | 36.67 |

## D.5 EFFICIENCY ANALYSIS

To demonstrate the influence of the number of prototypes in the dual-memory module on the computational complexity of PAMA, we analyze the relationships between the number of prototypes and the training time,

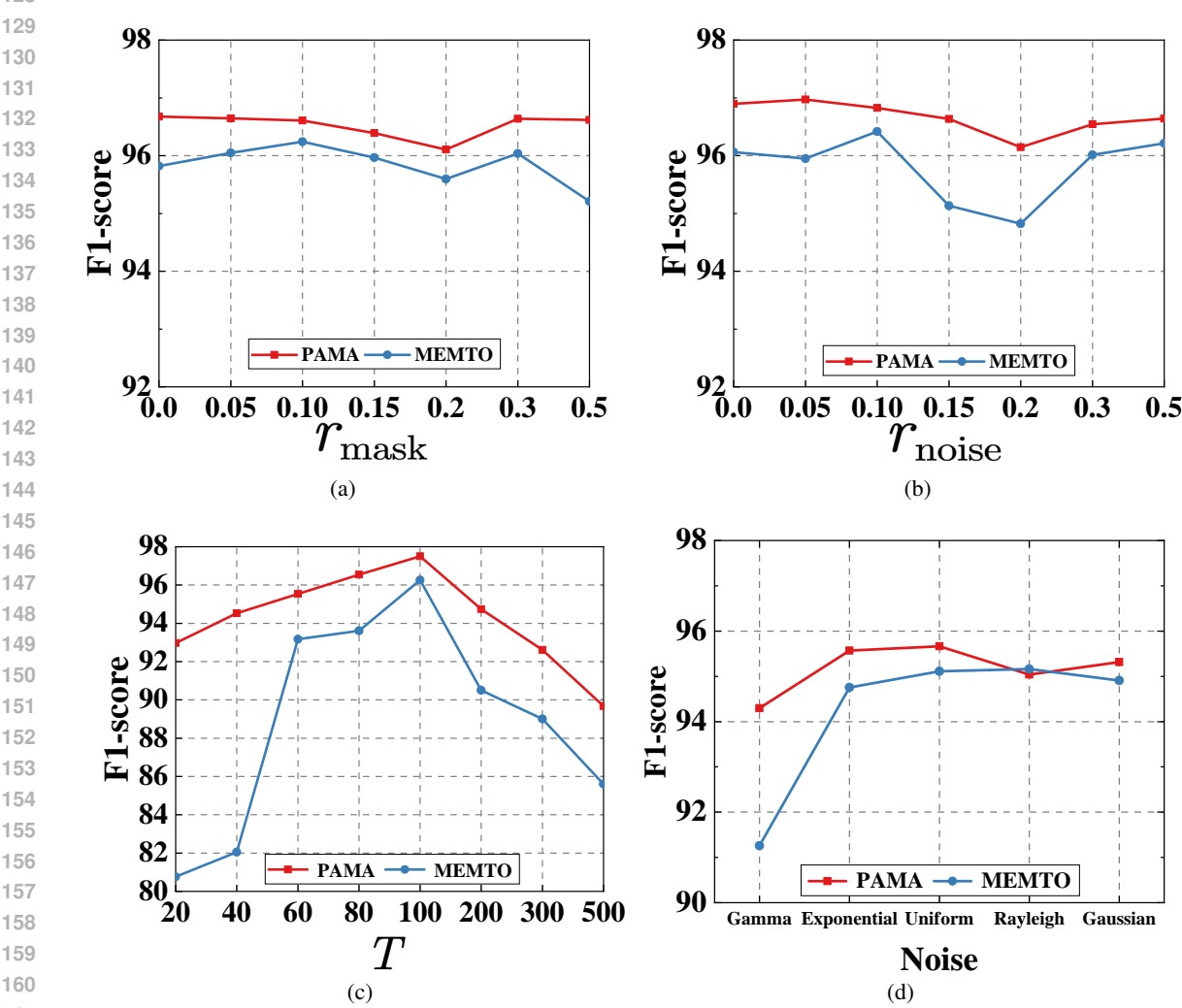

Figure 8: Average performance comparison of MEMTO and PAMA on the five datasets under different values of mask rates $r_{\mathrm{mask}}$, noise rates $r_{\mathrm{noise}}$, window lengths and different noise distributions.

Table 10: Computational complexity of PAMA and baseline model on three benchmark datasets in terms of training time (s/epoch), testing time (s/epoch)), and memory (GB).

| Datasets | MSL | | | SMAP | | | PSM | | |
|---|---|---|---|---|---|---|---|---|---|
| Complexity | MEMTO | H-PAD | PAMA | MEMTO | H-PAD | PAMA | MEMTO | H-PAD | PAMA |
| Training time(s) | 3.08 | 25.45 | 17.04 | 3.75 | 24.16 | 20.36 | 3.6 | 20.57 | 15.33 |
| Testing time(s) | 1.53 | 6.47 | 4.66 | 1.32 | 7.01 | 4.8 | 1.73 | 9.18 | 5.34 |
| Memory(GB) | 1.55 | 7.91 | 4.09 | 2.00 | 9.78 | 5.09 | 1.89 | 9.43 | 5.04 |

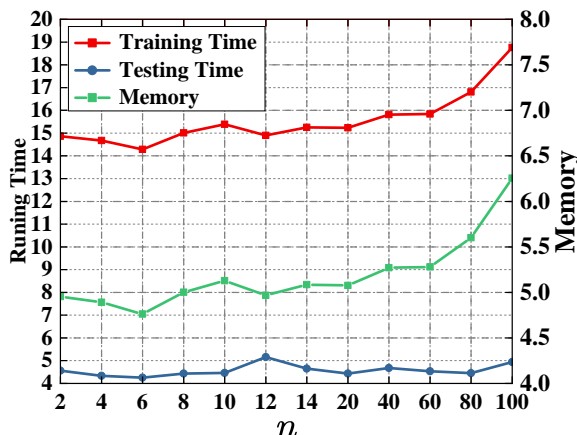

Figure 9: Influence of different number of prototypes on the computational complexity of PAMA.

testing time, and memory usage. The average training time, the average testing time and the average memory usage of PAMA on the three datasets are summarized in Fig. 9. It can be observed from Fig. 9 that as the number of prototypes increases, both the training time and memory usage increase, but the testing time remain relatively stable. Taking into account the results of Figs. 7-9, we set $n = 10$.

To observe the impact of dual-memory module or single-memory module on the computational complexity of the model, the differences between the training time, testing time and memory usage of MEMTO, H-PAD and PAMA are studied on the three datasets. The experimental results of MEMTO and PAMA are included in Table 10. One can observe from Table 10 that the computational complexity of PAMA is higher than that of MEMTO but lower than that of H-PAD. However, the detection performance of PAMA is relatively better than that of MEMTO and is close to that of H-PAD.

## D.6 VISUALIZATION ANALYSIS

To observe whether the pseudo anomalies generated by PAMA can cover most real anomalies or not, the t-SNE visualization results are shown in Fig. 10. Through the t-SNE visualization experiments on the input data and the generated pseudo-anomalies, Fig. 10 shows that the normal data demonstrate an obvious clustering structure, forming multiple dense normal clusters. Therefore, normal data possess strong inherent regularity and consistency in the representation space. Real anomalies are scattered between normal data and pseudo anomalies. Some anomalies are adjacent to normal data clusters, while others are mixed with pseudo anomalies, reflecting that real anomalies deviate from the normal pattern. Moreover, due to the complexity of real scenarios, these real anomalies present diverse distribution forms. The distribution of pseudo anomalies are relatively scattered and have no obvious clustering. In summary, the visualization experiments verify that pseudo anomalies generated based on prior knowledge are consistent with the actual distribution of anomalies in the real world.

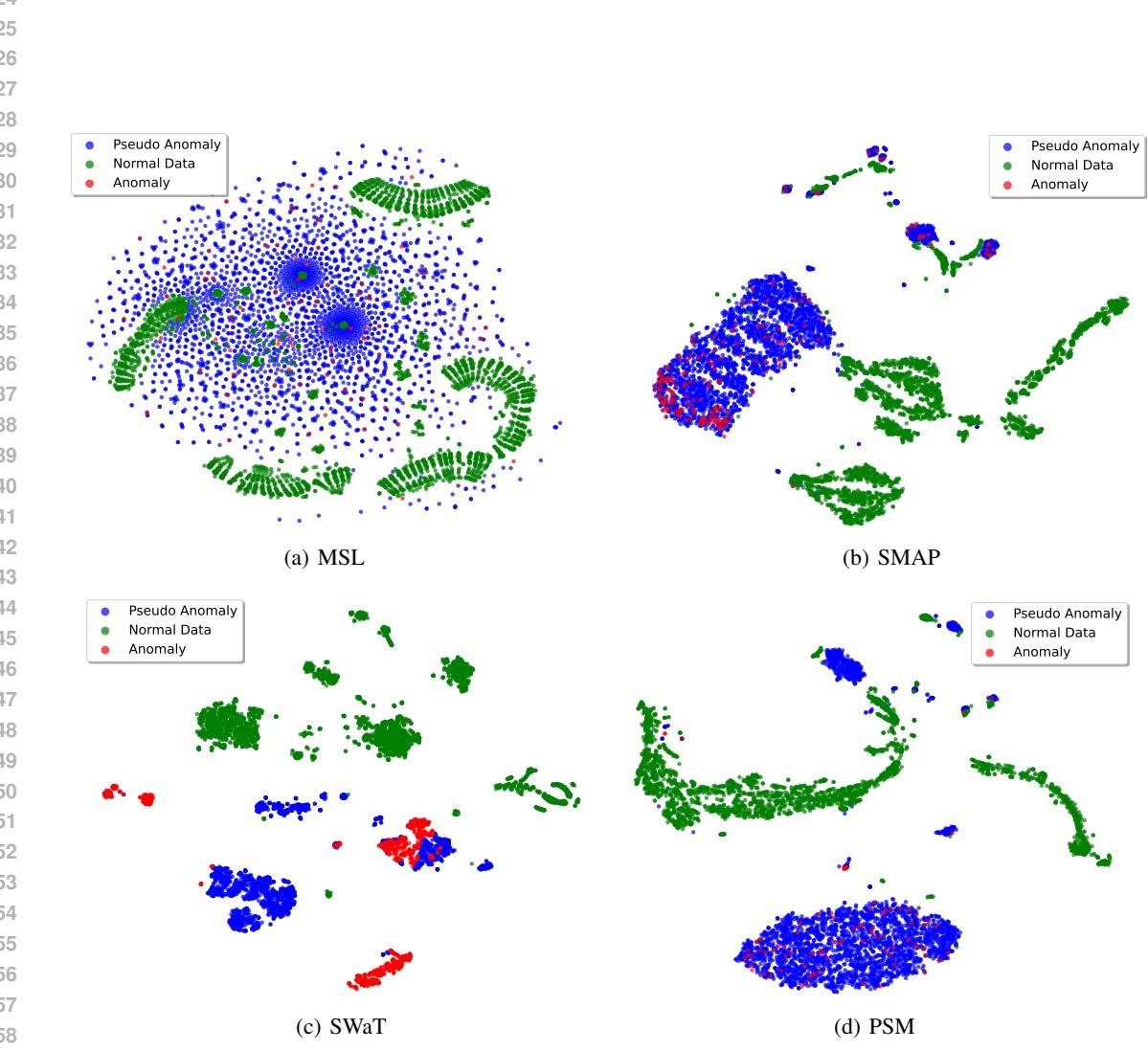

(a) MSL

(b) SMAP

(c) SWaT

(d) PSM

Figure 10: Visualization of t-SNE embeddings of original time series and pseudo anomalies on the four datasets. Green dots are the normal data, red are the anomalies, and blue are the pseudo anomalies generated by PAMA.

