# OpenReview forum: "PAMA: Dual-Memory Augmentation Assisted Pseudo-Anomaly Contrastive Learning for Multivariate Time Series Anomaly Detection"
_ICLR.cc/2026/Conference — Submitted to ICLR 2026_

### Official Review · Reviewer_Za4k · 2025-10-28

**Soundness:** 1
**Presentation:** 3
**Contribution:** 1
**Rating:** 4
**Confidence:** 4

**Summary:**

This paper aims to address the issue that time-series anomaly detection models tend to overgeneralize and become insensitive to anomalies. It proposes a method that artificially generates anomalies based on prior knowledge and employs contrastive learning to distinguish between normal and abnormal representations, thereby enhancing the model’s sensitivity to anomalies.

**Strengths:**

1. The paper is well-organized, logically structured, and easy to follow.

2. The authors conduct comparative experiments with baselines on multiple datasets, demonstrating some improvements in F1 score achieved by the proposed method.

**Weaknesses:**

1. Experimentally, the improvement in F1 score achieved by the proposed method is quite limited — on average, it does not exceed 0.5% compared to the best baseline.

2. In terms of methodological soundness, the paper lacks sufficient justification regarding whether the prior knowledge used for artificially generating anomalies is reasonable. Moreover, it is highly questionable whether adding random perturbations and scaling the maximum amplitude component in the spectrum can effectively cover all possible types of anomalies.

3. This approach, which relies on artificially generated anomalies and learning their representations, may ironically weaken the anomaly detection model’s ability to generalize — that is, to correctly identify unseen anomalies that were not observable during training.

4. The core problem the paper aims to address is the model’s overgeneralization and its inability to effectively capture anomalies. However, according to the experimental results, the proposed method does not show an improvement in recall under the threshold selection strategy adopted by the authors, which undermines the claim that the method can mitigate the overgeneralization issue.

**Questions:**

1. Could the authors provide a concrete justification or analysis demonstrating how the prior knowledge used to generate anomalies aligns with the actual distribution of anomalies in real-world applications?

2. Could the authors conduct additional experiments to verify that the proposed method indeed enhances anomaly sensitivity and mitigates model overgeneralization? For example, under the condition of achieving the best F1 score, could they compare the recall rates across different methods?

---

> ### Author Response · Authors · 2025-11-26
> **Rebuttal (part 1/N)：Response to weakness**
>
> >Experimentally, the improvement in F1 score achieved by the proposed method is quite limited — on average, it does not exceed 0.5% compared to the best baseline.
>
> **Response 1**: Thank you for raising the key issue. Compared with the baseline model MEMTO, PAMA has improved F1-score on average by more than 1.5%. Compared with the latest model H-PAD, PAMA outperformed the average level of H-PAD by 0.5% on F1-score. But the resource consumption of PAMA is much lower than that of H-PAD (shown as in the following table). Therefore, we believe that PAMA is still competitive with related methods.
>
> | Complexity Metric   |              MSL                 |             SMAP                   |                PSM                    |
> |---------------------|-------------------------|---------------------------|-------------------------|
> |                                 | H-PAD             PAMA   | H-PAD                PAMA     | H-PAD                     PAMA |
> | Training time(s)    | 25.45   / 17.04      | 24.16       /20.36       | 20.57      /15.33      |
> | Testing time(s)     | 6.47       /4.66       | 7.01        / 4.80         | 9.18        / 5.34       |
> | Memory(GB)          | 7.91       / 4.09       | 9.78         / 5.09        | 9.43       / 5.04       |
>
> >In terms of methodological soundness, the paper lacks sufficient justification regarding whether the prior knowledge used for artificially generating anomalies is reasonable. Moreover, it is highly questionable whether adding random perturbations and scaling the maximum amplitude component in the spectrum can effectively cover all possible types of anomalies.
>
> **Response 2**:
> Thanks a lot for your comment. We stated our response in two aspects:
> - For methodology soundness, pseudo anomalies and raw datasets are visualized together in feature space (shown in Fig. 10, Page 28). It is illustrated that pseudo anomalies can well depict the distribution of true anomalies, validating the reasonability of generating pseudo anomalies with prior knowledge.
> - For types of anomalies, PAMA has two branches to generate trend anomalies and seasonal anomalies respectively. Subsequent fusion of the two branches can generate global anomalies, point anomalies, and context anomalies. Ablation studies have been carried out to validating the necessary of the two branches (shown as in Table 7, Appendix D2).
>
> >This approach, which relies on artificially generated anomalies and learning their representations, may ironically weaken the anomaly detection model’s ability to generalize — that is, to correctly identify unseen anomalies that were not observable during training.
>
> **Response 3**: PAMA has two branches to artificially generate two types of pseudo anomalies which depict the two most prevalent categories of anomalies. Fusion of the two types of pseudo anomalies can cover other types of anomalies. Furthermore, learning pseudo-anomalous representations eliminates PAMA’s exposure to the problem of mistakenly judging noise as anomalies, enhancing PAMA’s generalization ability in identifying unseen anomalies.
>
> >The core problem the paper aims to address is the model’s overgeneralization and its inability to effectively capture anomalies. However, according to the experimental results, the proposed method does not show an improvement in recall under the threshold selection strategy adopted by the authors, which undermines the claim that the method can mitigate the overgeneralization issue.
>
> **Response 4**: There are several evaluation indicators for the improvement on the overgeneralization issue. As one of these indicators, Recall quantifies only the proportion of true positives. For comprehensive model evaluation, other indicators are considered to meet different requirements in real applications. We summarized numerical comparisons on AUC-ROC, AUC-PR, Aff-r and R-A-R as comprehensive evaluations of PAMA’s performance on five benchmark datasets. Comparison results concretely verify that PAMA can mitigate the overgeneralization issue.
>
>
> |Metric  | MSL               | SMAP              | PSM               | SWaT              | SMD               |
> |--------|------------|-------------------|-------------------|-------------------|-------------------|
> |          | MEMTO/ H-PAD/ PAMA| MEMTO/ H-PAD/ PAMA| MEMTO/ H-PAD/ PAMA| MEMTO/ H-PAD/ PAMA| MEMTO/ H-PAD/ PAMA|
> | AUC-ROC    | 49.99 / **58.67** / 50.68 | **59.59** / 59.13 / 59.09 | 49.75 / **69.15** / 51.38 | 45.41 / **67.93** / 64.90 | 73.24 / **79.43** / 55.53 |
> | AUC-PR     | 10.48 / 14.06 / **14.20** | 16.29 / 15.30 / **17.53** | 26.96 / **48.13** / 32.66| 11.45 / 14.05 / **17.86** | 10.35 / **51.92**/ 25.31 |
> | Aff-r      | 96.00 / 96.25 / **96.33** | 98.87 / 98.91 / **98.96** | 84.79 / 84.85 / **85.25** | **93.41** / 87.45 / 88.76 | 94.81 / 93.22 / **97.74** |
> | R-A-R      | 90.36 / 91.34 / **91.36** | 96.02 / 96.83 / **97.00** | 91.80 / 92.91 / **94.20** | 89.04 / 88.82 / 87.40 | 80.35 / 81.02 / **83.30** |

---

> ### Author Response · Authors · 2025-11-26
> **Rebuttal (part 2/N)：Response to question**
>
> >Could the authors provide a concrete justification or analysis demonstrating how the prior knowledge used to generate anomalies aligns with the actual distribution of anomalies in real-world applications?
>
> **Response 1**: Based on your valuable suggestion, we visualized distributions of generated pseudo anomalies and real anomalies in the feature space based on four benchmark datasets. As shown in Fig. 10, the feature distribution of pseudo anomalies exhibits a nearly-perfect overlap with the distribution of real anomalies on MSL, SMAP and PSM. It concretely verifies the prior knowledge of anomalies in time series, motivating the generation of pseudo anomalies.
>
> >Could the authors conduct additional experiments to verify that the proposed method indeed enhances anomaly sensitivity and mitigates model overgeneralization? For example, under the condition of achieving the best F1 score, could they compare the recall rates across different methods?
>
> **Response 2**: Experimentally, PAMA is parameterized to achieve the best F1 score on MSL and directly applied on the other datasets to produce all evaluation indicators. In this case, the recall rates still achieve suboptimal on almost benchmark datasets as shown in Table 1. However, PAMA gets the optimal performance with the other comprehensive evaluation indicators on most benchmark datasets. These results support numerical evidence for PAMA enhancing anomaly sensitivity and mitigating model overgeneralization.

---

> > ### Comment · Reviewer_Za4k · 2025-11-26
> >
> > Thank you for your response.
> >
> > **R1 in 1/N:**
> >
> > I list the F1 score improvement for each dataset and compute average improvement.
> >
> > |               | MSL   | SMAP  | PSM   | SMD   | SwaT  | Average |
> > | ------------- | ----- | ----- | ----- | ----- | ----- | ------- |
> > | Best Baseline | 95.45 | 97.21 | 99.12 | 96.17 | 95.09 | ---     |
> > | PAMA          | 96.10 | 97.42 | 99.11 | 95.53 | 94.84 | ---     |
> > | Improvement   | 0.65  | 0.21  | -0.01 | -0.64 | -0.25 | -0.008  |
> >
> > The improvement is very marginal and the average improvement is even negative.
> >
> > **R2 & R3 in 1/N:**
> >
> > The pseudo anomalies can not cover all the anomalies encountered in practical scenarios. For example, the anomalies located in the upper-left and lower-right regions of Fig. 10(c) cannot be well represented or covered by the pseudo anomalies. Thus, it is very likely that using the proposed method can degrade the performance of the model on unseen anomalies.
> >
> > **R2  in 2/N:**
> >
> > The F1 score only evaluate the performance of different models, but can not prove that the proposed method solve the problem of overgeneralization directly. The improvement in the F1 score may result from multiple factors and is not necessarily attributable solely to the mitigation of the over-generalization issue.
> >
> >
> >
> > Thus, my concern has not been solved and I will keep my score.

---

> > > ### Author Response · Authors · 2025-12-02
> > > **Rebuttal (part 3/N)：Response to reply**
> > >
> > > Thank you for your reply.
> > > >R1 in 1/N:The improvement is very marginal and the average improvement is even negative.
> > >
> > > **Response 1**:
> > > The advantages of PAMA are summarized in the following three aspects.
> > > - Across the five datasets in Table 1, PAMA demonstrates an average performance improvement of approximately 7% over U-Transformer.
> > > - Although the performance difference between PAMA and H-PAD is not significant, PAMA demonstrates significantly lower computational complexity (see Table 10). H-PAD is higher than PAMA in terms of training time, testing time and memory usage. Specifically, the average differences in training time, testing time and memory usage on the three datasets are respectively 5.82 seconds, 2.62 seconds and 4.30GB.
> > >
> > > | Difference | MSL | SMAP | PSM |
> > > |-------------------|---------|----------|---------|
> > > | Training time(s)  | 8.41    | 3.8      | 5.24   |
> > > | Testing time(s)   | 1.81    | 2.21     | 3.84   |
> > > | Memory(GB)     | 3.82   | 4.69     | 4.39    |
> > >
> > > - Experimental results in Tables 2, 4-5 and Figure 8 demonstrate that PAMA achieves substantial performance gains and significantly improved noise robustness.
> > >
> > > >R2 & R3 in 1/N:The pseudo anomalies can not cover all the anomalies encountered in practical scenarios. For example, the anomalies located in the upper-left and lower-right regions of Fig. 10(c) cannot be well represented or covered by the pseudo anomalies. Thus, it is very likely that using the proposed method can degrade the performance of the model on unseen anomalies.
> > >
> > > **Response 2**:
> > > The PAG module of PAMA is indeed unable to cover all the anomalies encountered in real scenarios. Cross-variable anomaly and other complex anomaly have been considered in our future work. The most important is that pseudo anomalies can be generated and included for anomaly detection. In Fig. 10 (c), some points in the blue (pseudo anomaly) and red (anomaly) areas located at the top left and bottom right corners overlap. This is because the number of pseudo anomalies generated in the corner areas in the current visualization experiments is relatively small, which is the direct cause of incomplete local coverage.
> > >
> > > >R2 in 2/N:The F1 score only evaluate the performance of different models, but can not prove that the proposed method solve the problem of overgeneralization directly. The improvement in the F1 score may result from multiple factors and is not necessarily attributable solely to the mitigation of the over-generalization issue.
> > >
> > > **Response 3**:
> > > PAMA not only has improved performance in F1 scores, but also shows significant performance improvements in indicators such as Aff-r, Range-AUC-ROC, and V-ROC. These indicators can measure the model's stable predictive ability and generalization ability on unseen data. Compared with H-PAD, PAMA demonstrates improvements of 1.75, 0.93 and 1.99 in terms of Aff- R, R-A-R and V-ROC, respectively.
> > > | Models      | Aff-r  | R-A-R  | V-ROC  |
> > > |------------------|--------|--------|--------|
> > > | Anomaly Trans    | 79.14  | 85.57  | 84.67  |
> > > | DCdetector       | 79.57  | 84.77  | 83.40  |
> > > | U-Transformer    | 53.24  | 68.42  | 68.38  |
> > > | MEMTO            | 80.54  | 85.73  | 85.06  |
> > > | H-PAD            | 89.57  | 86.69  | 85.56  |
> > > | PAMA             | **91.32**| **87.62**| **87.55**|
> > > | Improvement | **↑1.75**| **↑0.93**| **↑1.99**|

---

### Official Review · Reviewer_FXzj · 2025-10-29

**Soundness:** 3
**Presentation:** 3
**Contribution:** 2
**Rating:** 4
**Confidence:** 4

**Summary:**

The paper proposes PAMA, a multivariate time-series anomaly detection (MTSAD) framework that (i) synthesizes pseudo anomalies informed by prior knowledge, (ii) learns separate prototype memories for normal and anomalous patterns, and (iii) trains with a tailored contrastive objective to mitigate over-generalization when training data contain noise/anomalies. Pseudo-Anomaly Generation (PAG) perturbs trends (random linear drifts) and seasonality (scaling the dominant FFT frequency per channel) within randomly sampled windows to produce pseudo anomalies. Dual-Memory Augmentation (DMA) store prototypes; similarities between representations and prototypes (sigmoid of scaled dot-product) are used to produce memory-augmented features. For Pseudo-Anomaly Contrastive Learning (PACL), temporal (adjacent positives for normal, pseudo-anomaly negatives) and instance-level (cross-memory positives/negatives) losses are combined. Uncertainty Learning (UL) adds a KL-regularized variational head and a distance loss to stabilize normal representations; the decoder reconstructs from concatenated (UL-refined + original) features. Anomaly score multiplies latent distance to nearest normal prototype with reconstruction error.

**Strengths:**

- The division into PAG → DMA → PACL → UL is coherent; figures/equations formalize the pipeline and losses precisely.
- Separating normal vs. pseudo-anomalous prototype stores is a sensible mechanism to avoid “all-normal” memories that over-reconstruct anomalies.
- The temporal loss uses adjacency as positives and pseudo-anomalies as negatives; the instance loss leverages cross-memory augmentation—both are appropriate for time dependence and class asymmetry.

**Weaknesses:**

- The paper claims to “combine/utilize prior knowledge of anomalies” to generate pseudo anomalies, but in practice this “prior” is instantiated as generic, hand-crafted perturbations—adding random linear trends per channel and scaling the dominant FFT frequency in short windows—rather than knowledge derived from domain experts, causal structure, or external metadata. A more precise framing—and an empirical audit of how realistic these perturbations are for multivariate, cross-channel failure modes—would improve conceptual clarity.

- PAG perturbs the per-channel dominant frequency and injects linear trends. This may not capture multivariate cross-channel anomalies (e.g., lagged inter-sensor faults, topology-induced correlations) and could bias PACL toward single-channel spectral artifacts.

- Although the paper claims improved performance broadly, it remains unclear how PAMA handles anomalies with weak spectral signatures or non-stationary multi-sensor couplings (e.g., actuator-sensor loops) given the PAG design and memory size (10 prototypes/module) reported.

- The paper does not benchmark against CAROTS (ICML 2025)—which uses causality-preserving/disturbing augmentations and a similarity-filtered one-class contrastive loss—nor against CARLA (Pattern Recognition 2025)—which leverages contrastive learning with anomaly injection to sharpen the normal boundary. Because PAMA’s core claims also hinge on pseudo anomalies and contrastive objectives, comparison with these baselines are necessary to position its contribution relative to these state-of-the-art approaches.

- Please include F1 without point adjustment (and, ideally, the corresponding precision/recall) under the same thresholds and preprocessing per dataset. This will let readers assess robustness to the choice of point adjustment and quantify any inflation gap between adjusted vs. raw F1.

- Table 5 shows U-Transformer achieving strikingly high Aff-p on all datasets, and on SWaT, the performance of PAMA is notably lower than some baselines. Please provide a targeted discussion diagnosing these patterns.

- Figure 1’s caption should succinctly narrate the pipeline and define symbols. For Figures 3–4, reducing whitespace and margins and adding richer yet compact results would improve readability and interpretability.

**Questions:**

- PAG scales the dominant per-channel frequency and adds linear trends. How do you ensure these perturbations faithfully approximate multivariate anomalies where the signature is cross-channel phase/lag structure rather than single-channel spectra?

- What is the trade-off curve between prototype count and performance, and how does prototype collapse or redundancy get avoided during training?

- In the main text, the temporal/instance contrastive losses (Eqs. (7)–(9)) are written as raw dot products without a temperature or feature normalization term, whereas the appendix reports sensitivity to “temperature parameters. Does \tau refer to a contrastive temperature (as in InfoNCE), and if so, where is it inserted in Eqs. (7)–(9)?

---

> ### Author Response · Authors · 2025-11-28
> **Rebuttal (part 1/N)：Response to weakness**
>
> We sincerely appreciate your critical evaluation. Your comments and questions have been pivotal in identifying limitations and guiding critical enhancements to our model.
> >The paper claims to “combine/utilize prior knowledge of anomalies” to generate pseudo anomalies, but in practice this “prior” is instantiated as generic, hand-crafted perturbations—adding random linear trends per channel and scaling the dominant FFT frequency in short windows—rather than knowledge derived from domain experts, causal structure, or external metadata. A more precise framing—and an empirical audit of how realistic these perturbations are for multivariate, cross-channel failure modes—would improve conceptual clarity.
>
> **Response 1**:We stated the prior knowledge in three perspectives: (1) Trend anomaly and season anomaly are two common categories existed in time series, just as the meta features of anomalies in time series. (2) Trend and season pseudo anomalies are generated without including numerical statistics. (3) Visualization in the feature space on the benchmark datasets shows that pseudo anomalies can well depict the distribution of real anomalies. Therefore, PAMA can learn the most representative anomaly prototypes based on the prior knowledge. For multivariate and cross-channel failure modes, we have been endeavoring to tackle the issue but have not yet resolved it.
>
> >PAG perturbs the per-channel dominant frequency and injects linear trends. This may not capture multivariate cross-channel anomalies (e.g., lagged inter-sensor faults, topology-induced correlations) and could bias PACL toward single-channel spectral artifacts.
>
> **Response 2**: The pseudo anomaly generation process of PAMA is designed based on the assumption that variables are independent. For multivariate cross-channel anomalies, we plan to integrate the correlation matrix into the generation process in our future work.
>
> >Although the paper claims improved performance broadly, it remains unclear how PAMA handles anomalies with weak spectral signatures or non-stationary multi-sensor couplings (e.g., actuator-sensor loops) given the PAG design and memory size (10 prototypes/module) reported.
>
> **Response 3**: With the pseudo anomalies generated by PAG, a dual-memory module is designed to separately memorize normal prototypes and anomalous prototypes in the feature space. To well characterize normal and anomalous patterns, pseudo-anomaly contrastive learning is carried out on normal prototypes and anomalous prototypes. It is visualized in Fig. 10 that pseudo anomalies provide the most homogeneous characteristics with real anomalies. Therefore, the learned anomaly prototypes enable PAMA to distinguish anomalies from normal data.
>
> >The paper does not benchmark against CAROTS (ICML 2025)—which uses causality-preserving/disturbing augmentations and a similarity-filtered one-class contrastive loss—nor against CARLA (Pattern Recognition 2025)—which leverages contrastive learning with anomaly injection to sharpen the normal boundary. Because PAMA’s core claims also hinge on pseudo anomalies and contrastive objectives, comparison with these baselines are necessary to position its contribution relative to these state-of-the-art approaches.
>
> **Response 4**: In consideration of the baseline model MEMTO, PAMA is compared with H-PAD which is also designed based on MEMTO and achieves the best performance on benchmark datasets. For CARLA referred in your comments, the comparison results are briefly summarized in the original manuscript. However, we have problems in reproducing CAROTS to compare it with our model under identical experimental configurations. We have been struggling to settle these problem.
>
> | Datasets | Models | F1    | F1*   | AUC-ROC | AUC-PR  |
> |----------|--------|-------|-------|---------|---------|
> | MSL      | CARLA  | 96.07 | 52.27 | 50.20   | 13.50   |
> |          | PAMA   | **96.10** | **52.35** | 50.68   | **14.20** |
> | SMAP     | CARLA  | 96.97 | 52.92 | 54.20   | 14.85   |
> |          | PAMA   | **97.42** | **54.15** | 59.09   | **17.53** |
> | PSM      | CARLA  | 99.09 | **60.37** | 43.25   | 24.40   |
> |          | PAMA   | 99.11 | 59.35 | 51.38   | 32.66   |
> | SWaT     | CARLA  | 95.01 | 72.09 | **70.70** | **30.10** |
> |          | PAMA   | 94.84 | 75.19 | 64.90   | 17.86   |
> | SMD      | CARLA  | **95.66** | 51.14 | 45.41   | 15.10   |
> |          | PAMA   | 95.53 | 58.64 | 55.53   | 25.31   |
> | Average  | CARLA  | 96.56 | 57.76 | 52.71   | 19.59   |
> |          | PAMA   | **96.60** | **59.93** | 56.32   | 21.51   |

---

> ### Author Response · Authors · 2025-11-28
> **Rebuttal (part 2/N)：Response to weakness**
>
> >Please include F1 without point adjustment (and, ideally, the corresponding precision/recall) under the same thresholds and preprocessing per dataset. This will let readers assess robustness to the choice of point adjustment and quantify any inflation gap between adjusted vs. raw F1.
>
> **Response 5**: Thanks for your valuable suggestions. We supplemented further experiments to evaluate and quantify the impact of point adjustments on the performance of related models. Comparison results on five datasets are summarized in the following table (Table 9 in the revised manuscript). Note that F1* is the F1-score without point adjustments under the same thresholds and preprocessing per dataset. Regardless of point adjustments, PAMA promotes F1 scores greatly over the baseline model MEMTO and achieves the best F1 score on average.
>
> | Datasets | MEMTO| H-PAD  | CARLA | PAMA  |
> |----------|-------------------------|-------------------------|-------------------------|-------------------------|
> |     |    F1/F1* |F1/F1* | F1/F1*  | F1/F1*  |
> | MSL      | 94.36      / 22.70       | 95.45      / 50.11       | 96.07      /52.27       | **96.10**  / **52.35**   |
> | SMAP     | 96.44      / 27.23       | 97.21      /53.25       | 96.97     / 52.92       | **97.42** / **54.15**   |
> | PSM      | 98.18     / 52.15       | **99.12**  / 56.11       | 99.09      / **60.37**   | 99.11      / 59.35       |
> | SWaT     | 93.25      /29.01       | **95.09** / **76.07**   | 95.01      / 72.09       | 94.84      / 75.19       |
> | SMD      | 94.00     / 51.69       | 95.53      / **59.79**   | **95.66**  / 51.14       | 95.53      / 58.64       |
> | Average  | 95.24     /36.56       | 96.46      / 59.06       | 96.56      /57.76       | **96.60** /**59.93**   |
>
> >Table 5 shows U-Transformer achieving strikingly high Aff-p on all datasets, and on SWaT, the performance of PAMA is notably lower than some baselines. Please provide a targeted discussion diagnosing these patterns.
>
> **Response 6**: U-Transformer indeed achieves strikingly high Aff-p on all datasets as shown in Table 5. The SWaT dataset is a high-frequency time series data with strong correlations among variables. PAMA's dual memory enhancement and pseudo-anomaly contrastive learning mainly study the patterns among single variable. Among them, instance contrastive learning conducts contrastive learning on the series obtained from different memory-augmention of single-channel time series, so it mainly studies the normal and anomalous patterns of single-channel series. The learning of inter-channel correlations is not as good as that of other methods. This is a limitation of PAMA, and we will make improvements in this aspect in the future.
>
> >Figure 1’s caption should succinctly narrate the pipeline and define symbols. For Figures 3–4, reducing whitespace and margins and adding richer yet compact results would improve readability and interpretability.
>
> **Response 7**: Thanks a lot for suggestions on paper organizing. We have refined through the paper.

---

> ### Author Response · Authors · 2025-11-28
> **Rebuttal (part 3/N)：Response to question**
>
> >PAG scales the dominant per-channel frequency and adds linear trends. How do you ensure these perturbations faithfully approximate multivariate anomalies where the signature is cross-channel phase/lag structure rather than single-channel spectra?
>
> **Response 1**: PAMA assumes that variables are independent to each other such that pseudo anomalies generated by PAD can well approximate anomalies on each single variable, but not on multivariate anomalies. We have been devoting ourselves to tackling the issue.
>
> >What is the trade-off curve between prototype count and performance, and how does prototype collapse or redundancy get avoided during training?
>
> **Response 2**: Please refer Fig. 7(c) for the trade-off curve between prototype numbers and F1 scores. For prototype collapse and redundancy, uncertainty learning is designed to make training process adaptable to avoid prototype collapse and to reduce redundancy.
>
> >In the main text, the temporal/instance contrastive losses (Eqs. (7)–(9)) are written as raw dot products without a temperature or feature normalization term, whereas the appendix reports sensitivity to “temperature parameters. Does \tau refer to a contrastive temperature (as in InfoNCE), and if so, where is it inserted in Eqs. (7)–(9)?
>
> **Response 3**: The temperature parameter is not a parameter of temporal/instance contrastive loss. The sensitivity of $\tau$ studies in Fig. 7(b) is a key parameter of GL-Encoder. We specified the parameter description in Section B. 2 (Page 15).

---

### Official Review · Reviewer_dJLQ · 2025-10-30

**Soundness:** 3
**Presentation:** 2
**Contribution:** 2
**Rating:** 4
**Confidence:** 2

**Summary:**

This paper proposes a method that (i) synthesizes pseudo-anomalies, (ii) stores normal and pseudo-anomalous patterns in separate memories, (iii) drives their representations apart, and (iv) leverages both normal and anomalous patterns during training. Across multiple baselines, the average F1 score is notably strong.

**Strengths:**

- The authors revisit the common assumption that training data are nearly anomaly-free and address its limitations by introducing a dual-memory design, one for normal patterns and one for pseudo-anomalies, for detection. This represents a novel direction relative to prior work.
- As the ablation studies show, combining PAMA's components yields clear, quantitative performance gains.

**Weaknesses:**

- The Abstract states that *"Most methods assume that training data are clean and ignore the characteristics of anomalous data,"* yet it remains unclear how PAMA concretely resolves this in practice. Although the numerical experiments use standard time‑series anomaly‑detection benchmarks, direct evidence that the stated challenge is specifically addressed would be helpful.
- While reexamining the *"few anomalies in training data assumption"* is novel, the dual-memory module appears structurally similar to prior work, leaving the degree of methodological novelty somewhat unclear. Please clarify the precise differences between CutAddPaste (with PAG) and MEMTO (with a dual-memory module).
- It is difficult to assess the statistical significance of the results in Tables 1 and 2. Because the proposed method introduces several hyperparameters, the robustness of the reported gains to these choices is not yet evident.
- Relative to MEMTO, training time and memory consumption are substantially higher.

**Questions:**

- Please state clearly how CutAddPaste + PAG differs from MEMTO + dual memory, both conceptually and operationally.
- Is the approach restricted to pseudo-anomalies based on trend and seasonality, or does it generalize to other anomaly types?
- How sensitive is performance to the entropy loss, i.e., how much does it change when this term is removed?

---

> ### Author Response · Authors · 2025-11-23
> **Rebuttal (part 1/N)：Response to weakness**
>
> We greatly appreciate the suggestions and questions you have provided. Below, we address each of your points one by one.
> >The Abstract states that "Most methods assume that training data are clean and ignore the characteristics of anomalous data," yet it remains unclear how PAMA concretely resolves this in practice. Although the numerical experiments use standard time‑series anomaly‑detection benchmarks, direct evidence that the stated challenge is specifically addressed would be helpful.
>
> **Response 1**: In accordance with previous methods, PAMA also assumes that the training data is clean. However, different from previous methods, PAMA takes the advantages of prior knowledge of anomaly characteristics (such as season anomalies and anomalous trend) to generate pseudo anomalies, learn pseudo-anomalous prototypes, and conduct pseudo-anomaly contrastive learning. It is very important to learn the most representative features of real anomalies to identify unseen anomalies. On the one hand, the distributions of pseudo anomalies and real anomalies in benchmark datasets have been visualized in Fig. 10 (Appendix D. 6), illustrating the effectiveness of depicting real anomalies with pseudo anomalies. On the other hand, the detection performance is enhanced with the assistance of pseudo anomalies generated with prior knowledge of anomaly characteristics, numerically illustrated in Table 6. Both visualization and numerical results are direct evidence for PAMA concretely solving the issue of ''ignoring the characteristics of anomalous data''.
> >While reexamining the "few anomalies in training data assumption" is novel, the dual-memory module appears structurally similar to prior work, leaving the degree of methodological novelty somewhat unclear. Please clarify the precise differences between CutAddPaste (with PAG) and MEMTO (with a dual-memory module).
>
> **Response 2**: We clarified the specific differences between the pseudo-anomaly generation module in PAMA and CutAddPaste, as well as the memory module between PAMA and MEMTO. Differences are summarized in the following tables.
> |     | PAMA                                                                       | CutAddPaste                          |
> |-------------------------|----------------------------------------------------------------------------|-------------------------------------|
> | Pseudo Anomaly Types    | &nbsp;&nbsp;Trend anomaly, Seasonal anomaly, Point anomaly, Global anomaly | Trend anomaly, Point anomaly        |
> |Augmentation Strategy   | &nbsp;&nbsp;Trend perturbation and seasonal perturbation                   | Trend perturbation                  |
> |Functional positioning  | &nbsp;&nbsp;Serve as the pseudo-anomalous data source, and train the memory module to obtain pseudo-anomalous prototypes. | Train the classifier                |
>
> |      | PAMA                                                          | MEMTO                          |
> |----------------------------|---------------------------------------------------------------|--------------------------------|
> | Data Source                | Input data,  Pseudo-anomalous data                  | Input data                 |
> | Memory Module    | Normal memory module, Pseudo-anomalous memory module       | Normal memory module           |

---

> ### Author Response · Authors · 2025-11-23
> **Rebuttal (part 2/N)：Response to weakness**
>
> >It is difficult to assess the statistical significance of the results in Tables 1 and 2. Because the proposed method introduces several hyperparameters, the robustness of the reported gains to these choices is not yet evident.
>
> **Response 3**: Experimentally, PAMA uses identical parameters to MEMTO (the baseline method) and solely adjusts its additional parameters. As shown in the following table, PAMA outperforms the latest results on most datasets. Furthermore, the performance of PAMA is promoted by 1.4% on average in comparison with the baseline MEMTO.
> | Model  | MSL (Pre/Rec/F1) | SMAP (Pre/Rec/F1) | PSM (Pre/Rec/F1) | SMD (Pre/Rec/F1) | SWaT (Pre/Rec/F1) | Avg F1  |
> |--------|------------------|-------------------|------------------|------------------|-------------------|---------|
> | MEMTO  | 92.07/96.76/94.36| 93.75/99.29/96.44 | 97.44/98.96/98.18| 92.89/95.13/94.00| 89.13/97.76/93.25 | 95.24   |
> | H-PAD  | 94.05/96.88/95.45| 96.00/98.45/97.21 | 98.82/99.41/**99.12**| 92.86/98.20/95.45| 94.14/98.89/**95.09** | 96.46   |
> | PAMA   | 94.34/97.93/**96.10**| 95.95/98.95/**97.42**| 98.86/99.35/99.11| 92.05/99.29/**95.53**| 92.44/97.36/94.84 | **96.60** |
>
> >Relative to MEMTO, training time and memory consumption are substantially higher.
>
> **Response 4**: As shown in the table below, compared with MEMTO, PAMA increases training time and memory consumption, but significantly improves model performance. Compared with H-PAD, which is also based on the memory prototype, although the performance improvement of PAMA is relatively small, the training time and memory consumption are greatly reduced. Considering both the improvement of model performance and resource consumption, it indicates that introducing dual memory modules is a good choice.
> | Complexity Metric   | MSL                  | SMAP                 | PSM                  |
> |---------------------|----------------------|----------------------|----------------------|
> |                     | MEMTO / H-PAD  /PAMA   | MEMTO  /H-PAD  /PAMA   | MEMTO / H-PAD  /PAMA   |
> | Training time(s)    | 3.08   / 25.45 / 17.04  | 3.75 /  24.16 / 20.36  | 3.6  /  20.57  / 15.33  |
> | Testing time(s)     | 1.53  /  6.47 / 4.66   | 1.32  /  7.01  /  4.8    | 1.73 /  9.18 / 5.34   |
> | Memory(GB)          | 1.55  / 7.91 / 5.09   | 2.00  /  9.78 /  5.09   | 1.89 /  9.43  /  5.04   |

---

> > ### Comment · Reviewer_dJLQ · 2025-11-27
> > **Reply**
> >
> > I thank the authors for their detailed rebuttal, clarifications, and additional experiments.
> > However, my main concerns remain: the methodological novelty still appears incremental, largely combining and extending existing ideas, and the evidence for statistically significant, robust gains over strong baselines is limited, so the complexity–performance trade-off is not clearly favorable. **In particular, improvements on the order of 1.x% are difficult to interpret without error bars, confidence intervals, or multi-seed variance, so the claimed advantage over the baselines is not fully convincing.** Moreover, the current pseudo-anomaly design has limited generality (e.g., it cannot handle variable-related anomalies), and some reported improvements (such as those from the entropy loss) are quite small. For these reasons, while I appreciate the author's efforts and view the work as a modest contribution to this line of research, I do not find sufficient justification to raise my original score and will keep the score and confidence unchanged.

---

> > > ### Author Response · Authors · 2025-12-02
> > > **Rebuttal (part 4/N)：Response to reply**
> > >
> > > Thank you for your reply.
> > >
> > > The two core innovations of our proposed PAMA can be summarized as the following two aspects.
> > > - **Dual-memory augmentation (DMA)** performs data augmentation through the normal and pseudo-anomalous memory modules to get memory-augmented representations. **DMA** can effectively enhance the learning ability for prototype features by performing augmentation processing on different representations. (c.f. Section 3.2).
> > > - **Pseudo-anomaly contrastive learning (PACL)** performs temporal contrastive learning and instance contrastive learning to better utilize normal and anomalous patterns. (c.f. Section 3.3).
> > > The greatest innovation of PAMA lies in learning pseudo-anomalous prototypes and performing dual-memory augmentation through **DMA**, and conducting contrastive learning of memory-augmented representations through **PACL**.

---

> ### Author Response · Authors · 2025-11-23
> **Rebuttal (part 3/N)：Response to question**
>
> We greatly appreciate the suggestions and questions you have provided. Below, we address each of your points one by one.
>
> >Please state clearly how CutAddPaste + PAG differs from MEMTO + dual memory, both conceptually and operationally.
>
> **Response 1**:
> - The innovative idea of the PAG in PAMA is inspired by CutAddPaste. PAG adopts two branches, one for seasonal perturbation of the original series and the other for trend perturbation of the sampled series. Finally, the seasonal perturbation series and the trend perturbation series are fused to obtain the final pseudo anomaly.
> - PAMA differs from MEMTO in two aspects. On the one hand, PAMA deals with normal data and pseudo-anomalies while MEMTO deals with only normal data. On the other hand, PAMA is designed with a dual memory module to memory normal prototypes and the pseudo-anomalous prototypes respectively, while MEMTO has only one memory bank for normal prototypes.
>
> >Is the approach restricted to pseudo-anomalies based on trend and seasonality, or does it generalize to other anomaly types?
>
> **Response 2**: PAMA has two branches to generate trend anomalies and seasonal anomalies respectively. Adaptive fusion of the two branches can generate global anomaly, point anomaly, and context anomalies. However, the limitation of PAMA is that it cannot generate variable-related anomaly. We will make improvements to this in our future work.
>
> >How sensitive is performance to the entropy loss, i.e., how much does it change when this term is removed?
>
> **Response 3**: Since the baseline model MEMTO includes entropy loss, we retained it in PAMA. Based on your question, we conducted the corresponding ablation study and listed the results in the following table. It can be deduced from the table that compared to no entropy loss, PAMA with entropy loss achieves a 0.24% average improvement across four datasets.
>
> | Model    | MSL (Pre/Rec/F1)       | SMAP (Pre/Rec/F1)      | PSM (Pre/Rec/F1)       | SWaT (Pre/Rec/F1)      |
> |---------------------|-------------------------|-------------------------|-------------------------|-------------------------|
> | w/o${L}_{\text{entr}}$| 94.12/97.84/96.01       | 95.81/98.81/97.21       | 98.65/99.14/98.78       | 92.01/99.04/95.21       |
> | ${L}_{\text{entr}}$| 94.34/97.93/**96.10**   | 95.95/98.95/**97.42**   | 98.86/99.35/**99.11**   | 92.05/99.29/**95.53**   |

---

### Official Review · Reviewer_WtDd · 2025-10-31

**Soundness:** 3
**Presentation:** 2
**Contribution:** 3
**Rating:** 6
**Confidence:** 3

**Summary:**

This paper introduces PAMA (Dual-Memory Augmentation Assisted Pseudo-Anomaly Contrastive Learning), a novel method for multivariate time series anomaly detection (MTSAD). The main contributions of the paper are:

(1) A modified Pseudo-Anomaly Generation (PAG) module that creates realistic pseudo-anomalies by introducing both trend and seasonal perturbations to the original time series.

(2) A Dual-Memory Module (DMM) that uses two separate memory banks to store prototypes for normal and pseudo-anomalous patterns, respectively.

(3) A Dual-Memory Augmentation (DMA) mechanism that leverages these distinct memory modules to perform data augmentation on the feature representations.

(4) A Pseudo-Anomaly Contrastive Learning (PACL) framework that applies both temporal and instance-level contrastive objectives to the memory-augmented representations.

Experimental results show that PAMA outperforms 13 relevant baselines on 5 benchmark datasets.

**Strengths:**

(1) The core contribution, the dual-memory architecture for explicitly storing and utilizing both normal and pseudo-anomalous prototypes, is novel and well-motivated. It provides a principled way to leverage generated anomalies beyond just being negative samples in a contrastive loss.

(2) The paper presents extensive experiments on 7 datasets against several baselines. Moreover, the ablation studies and parameter analyses provide strong empirical support for the proposed architecture.

(3) Overall, the paper is clearly written. The motivation is compelling and the experimental findings are presented effectively.

**Weaknesses:**

(1) PAMA involves a large number of hyperparameters that require tuning. The sensitivity analysis shows that performance can fluctuate significantly with changes in these values, particularly on the MSL dataset. This suggests that the model may require careful and extensive tuning for each new dataset, which could be a practical limitation.

(2) Figures 3 and 4 are inconsistent with the main experimental setup, which is reported on 5 datasets. Figure 3 excludes SMD and Figure 4 excludes both SMD and SWaT, without any explanation. It is recommended to include all 5 datasets in both figures for completeness, as the space does not appear to be a limiting factor.

(3) There are some minor issues with the writing and presentation quality of the paper.
- The presentation of related work in the introduction contains a confusing chronological error. After discussing methods from 2023 and 2024, the paper incorrectly frames a 2018 paper as a "subsequent study", which weakens the logical flow.
- In the last sentence of Related Work (Lines 121-122), the authors claim that pseudo-anomalies from prior work are "significantly different from the real anomalies" without providing any supporting evidence. This part would be stronger if the authors briefly explained why.

**Questions:**

(1) The strongest baseline, H-PAD, is missing in Table 2. Can you provide the AUC-ROC / AUC-PR results for H-PAD as well?

(2) In Lines 331-333, it says PAMA significantly improves over CAE-AD on MSL, SMAP, and SMD. However, the results of CAE-AD for SMD are not reported in Table 1. Is "SMD" a typo of "SWaT"?

(3) The submission currently lacks an anonymous repository link and does not include any supplementary material. Do the authors plan to make the implementation code public to ensure the reproducibility of this work?

---

> ### Author Response · Authors · 2025-11-20
> **Rebuttal (part 1/N)：Response to weakness**
>
> We greatly appreciate the suggestions and questions you have provided. Below, we address each of your points one by one.
> >PAMA involves a large number of hyperparameters that require tuning. The sensitivity analysis shows that performance can fluctuate significantly with changes in these values, particularly on the MSL dataset. This suggests that the model may require careful and extensive tuning for each new dataset, which could be a practical limitation.
>
> **Response**: Regarding the hyperparameters $\lambda_1$, $d_{model}$, $\tau$, $n$, PAMA adopts identical parameter settings to the baseline model MEMTO. To further explore their influence, we conducted parameter sensitivity experiments. Regarding the newly proposed hyperparameters in PAMA, namely $\alpha$, $r_{s}$, $\lambda_2$, $\lambda_3$ and $\lambda_4$, they are necessarily adjusted for the optimal performance on MSL. First, $\alpha$ and $r_{s}$ are adjusted by tuning one while fixing another. Then, with the optimal $\alpha$ and $r_{s}$, we carried out grid search for optimal $\lambda_2$, $\lambda_3$ and $\lambda_4$. These optimal parameters in MSL are applied directly on the other datasets without further adaption, as well as the other unseen dataset.
>
> >Figures 3 and 4 are inconsistent with the main experimental setup, which is reported on 5 datasets. Figure 3 excludes SMD and Figure 4 excludes both SMD and SWaT, without any explanation. It is recommended to include all 5 datasets in both figures for completeness, as the space does not appear to be a limiting factor.
>
> **Response**: Based on your valuable suggestions, we have supplemented experimental results on SMD and SWaT in Figs. 3 and 4, followed with result analysis. As shown in Fig. 3, each module enhances the detecting performance on SMD. In Fig. 4 (a) for $\alpha$ perturbation, F1-score remains relatively stable on SMD and SWaT and also achieves the best on $\alpha=1.1$. For the sampling ratio $r_{s}$ in Fig. 4 (b), F1-scores on SMD and SWaT show significant downward trends after $r_s=0.2$.
>
> >The presentation of related work in the introduction contains a confusing chronological error. After discussing methods from 2023 and 2024, the paper incorrectly frames a 2018 paper as a "subsequent study", which weakens the logical flow.
>
> **Response**: In the Introduction section, we have revised the related work on the issue that ''there is noise and anomalous data interference during the training process, and some researchers proposed methods based on anomaly assumption.''
>
> >In the last sentence of Related Work (Lines 121-122), the authors claim that pseudo-anomalies from prior work are "significantly different from the real anomalies" without providing any supporting evidence. This part would be stronger if the authors briefly explained why.
>
> **Response**: In the Related Work section, previous methods, such as CutAddPaste, merely disrupts local sequences, ignoring the inherent seasonality and long-term trends of time series, resulting in the generation of only simple point anomalies and local trend anomalies. Therefore, we claim that ''the pseudo-anomalies in previous work are significantly different from the real anomalies.'' As your valuable suggestions, we briefly explained the reasons in the Related Work section and supplemented the visualization experiment in Fig. 10 for supporting evidence.

---

> ### Author Response · Authors · 2025-11-20
> **Rebuttal (part 2/N)：Response to question**
>
> We greatly appreciate the suggestions and questions you have provided. Below, we address each of your points one by one.
>
> >The strongest baseline, H-PAD, is missing in Table 2. Can you provide the AUC-ROC / AUC-PR results for H-PAD as well?
>
> **Response**: As your valuable suggestions, we supplemented the AUC-ROC/ AUC-PR experimental results of H-PAD in Table 2. We briefly listed the results of H-PAD for quick comparison with our model.
> | Criteria/Datasets | MSL         | SMAP        | PSM         | SWaT        | SMD         | Avg.        |
> |---------------|-------------|-------------|-------------|-------------|-------------|-------------|
> | AR            |             |             |             |             |             |             |
> | H-PAD         | **58.67**   | 59.13 | **69.15**   | 67.93| **79.43**   | **66.86**   |
> | PAMA          | 50.68 | 59.09       | 51.38| 64.9        | 55.53       | 56.32 |
> | AP            |             |             |             |             |             |             |
> | H-PAD         |14.06 | 15.30       | **48.13**   | 14.05       | **51.92**   | **28.69**   |
> | PAMA          | **14.20**   | **17.53**   | 32.66 |17.86 | 25.31       | 21.51|
>
> >In Lines 331-333, it says PAMA significantly improves over CAE-AD on MSL, SMAP, and SMD. However, the results of CAE-AD for SMD are not reported in Table 1. Is "SMD" a typo of "SWaT"?
>
> **Response**: Yes. It is indeed a typo. We have corrected ‘SMD’ as ‘SWaT’ in lines 344-345 and checked through the whole paper to confirm writing correctness.
>
> >The submission currently lacks an anonymous repository link and does not include any supplementary material. Do the authors plan to make the implementation code public to ensure the reproducibility of this work?
>
> **Response**: Actually, we have uploaded the code to Github. The source code is available at https://github.com/Qiantang-net/PAMA. The website will appear in the Camera-Ready version for easy reproduction if our paper is accepted.

---

### Meta-Review · Area_Chair_hfxR · 2026-01-06

**Summary:**

The paper proposes PAMA for multivariate time series anomaly detection, combining (i) pseudo-anomaly generation (trend/seasonality perturbations), (ii) dual memory banks for normal vs. pseudo-anomalous prototypes, (iii) dual-memory augmentation, and (iv) pseudo-anomaly contrastive learning (temporal + instance losses), with an uncertainty-learning component. The motivation—mitigating overgeneralization and explicitly leveraging anomaly characteristics—is reasonable, and the experimental section is broad (many baselines, multiple datasets, ablations).

However, the discussion converges on two decisive issues that remain unresolved after rebuttal:
- The empirical advantage over strong baselines is not consistently convincing, and in some key comparisons the gains are marginal, inconsistent, or even negative depending on the metric and dataset. In particular, when compared to the strongest baselines (e.g., H-PAD), PAMA’s F1 improvements are small and mixed across datasets, and the reliance on “point-adjusted” F1 further complicates interpretation. Multiple reviewers explicitly request stronger statistical validation, and one reviewer maintained a below-threshold score after rebuttal specifically because the improvements are hard to interpret without error bars.
- The pseudo-anomaly “prior knowledge” is largely heuristic and single-channel, with acknowledged limitations for cross-variable / inter-channel anomalies, which are central in multivariate anomaly detection. Several reviewers argue this weakens the paper’s claims about general anomaly coverage and may risk overfitting to the generated perturbation family. The authors ultimately concede that cross-variable anomalies are not handled and are left to future work—this undermines the generality of the core mechanism and the strength of the “prior knowledge” framing.

Although the rebuttal adds useful clarifications, the remaining concerns go to the heart of the paper’s novelty/impact and the credibility of the performance claims. As such, the submission does not yet meet the acceptance bar.

**Reviewer Concerns:**

> A. Baseline coverage and fairness of comparisons

**Addressed**

- The authors added missing baseline results (e.g., H-PAD AUC-ROC/AUC-PR in Table 2) and clarified/corrected at least one reported typo (SMD vs SWaT).
- They provided additional comparison tables including CARLA under similar metrics, and supplied a complexity comparison vs MEMTO/H-PAD.

**Outstanding**
- Multiple reviewers note missing or insufficient positioning against very recent contrastive/pseudo-anomaly methods (e.g., CAROTS), where the paper’s core claims overlap conceptually. The authors state they had reproduction issues and therefore could not include CAROTS under identical configurations. This leaves the method’s novelty and empirical standing relative to current SOTA ambiguous.
- In at least one reviewer’s reading, the strongest-baseline comparison on a per-dataset basis does not show a consistent win for PAMA (some datasets show regressions). The rebuttal’s emphasis on being better than weaker baselines (e.g., U-Transformer) does not resolve the key “best competitor” question.

> B. Statistical significance, robustness, and interpretability of “small” gains

**Addressed**

- The authors expanded parameter sensitivity and ablations, and provided additional metrics beyond point-adjusted F1 (e.g., AUC-ROC/AUC-PR, Aff-r, R-A-R, V-ROC), arguing these relate to robustness/generalization.

- They claim that PAMA largely reuses MEMTO’s hyperparameters and only tunes additional ones (and sometimes only tunes on one dataset and transfers to others).

**Outstanding**

- The central critique remains: improvements over strong baselines are small and hard to trust without uncertainty estimates (error bars, confidence intervals, multi-seed variance). One reviewer explicitly maintained their score, emphasizing that ~1% gains (or smaller) are not persuasive without statistical validation.
- The rebuttal adds more metrics, but this does not substitute for variance-aware evidence. Moreover, the metric picture is mixed: for example, the AUC-ROC/AUC-PR table included in the response shows H-PAD outperforming PAMA on several datasets, which complicates claims of consistent superiority.
- Some improvements (e.g., entropy loss contribution ~0.24% average) are acknowledged as quite small, reinforcing the perception that parts of the method add complexity with limited measurable benefit.

> C. Validity and generality of pseudo-anomaly generation (“prior knowledge”)

**Addressed**

- The authors clarified that their “prior knowledge” refers to broad anomaly meta-patterns and added a feature-space visualization intended to show pseudo anomalies overlap with real anomalies.
- They clarified scope: PAMA can generate several anomaly types via fusion, and explicitly acknowledged limitations.

**Outstanding**

- Reviewers argue that “prior knowledge” is hand-crafted perturbations rather than domain- or causality-grounded knowledge, and may not approximate realistic multivariate failure modes. The authors effectively concede this limitation and defer cross-variable anomaly modeling to future work, which is a significant weakness for multivariate anomaly detection.
- The visualization evidence is contested: one reviewer points out regions where pseudo anomalies do not cover real anomalies well and argues this may harm generalization to unseen anomaly types. The authors’ response does not fully resolve the concern that the generation family may be systematically mismatched to certain anomaly modes.

> D. Claims about mitigating overgeneralization and improving anomaly sensitivity

**Addressed**

- The authors argue that improvements in F1 plus additional robustness-like metrics provide evidence of enhanced sensitivity and generalization.
- They also provided F1 without point adjustment (F1*) to quantify inflation from point adjustment and to show that conclusions are not solely driven by adjustment.

**Outstanding**

- Reviewers highlight that F1 alone does not prove reduced overgeneralization, and the mechanism should ideally be supported by targeted analyses. The responses largely reframe the argument in terms of additional metrics, but do not directly demonstrate that overgeneralization is mitigated in a causal/diagnostic sense.
- The paper’s own explanations acknowledge that recall does not consistently improve under their thresholding strategy; this weakens the narrative that the method systematically increases anomaly sensitivity.

**Reviewer Scores:**

- `WtDd (6, borderline accept): likely remains 6`. The rebuttal addressed missing H-PAD results, figure completeness, and writing issues, but the added AUC tables and the broader discussion reveal that “best baseline” superiority is not uniform, so a strong upward shift is unlikely.
- `dJLQ (4, borderline reject): likely remains 4.` This reviewer explicitly states that novelty is incremental and the lack of error bars/multi-seed variance makes ~1% gains unconvincing; the rebuttal did not provide the requested statistical validation.
- `FXzj (4, borderline reject): likely stays 4.`. The authors added F1 without point adjustment and included CARLA comparisons and discussion about inter-channel correlation limitations, but they still concede that PAG is essentially single-variable and do not add CAROTS or stronger multi-channel realism audits.
- `Za4k (4, borderline reject): likely remains 4.` This reviewer directly computed per-dataset improvements vs best baseline and found the average improvement negative. They were unconvinced by the pseudo-anomaly coverage argument and maintained the score.

Overall, even with full discussion, the balance of scores likely stays below the threshold because the core “why this is reliably better” question remains unresolved.

---

### Decision · Program_Chairs · 2026-01-26

Reject